# Can Recommender Systems Teach Themselves? A Recursive Self-Improving Framework with Fidelity Control

## Abstract

The scarcity of high-quality training data presents a fundamental bottleneck to scaling machine learning models. This challenge is particularly acute in recommendation systems, where extreme sparsity in user interactions leads to rugged optimization landscapes and poor generalization. We propose the Recursive Self-Improving Recommendation (RSIR) framework, a paradigm in which a model bootstraps its own performance without reliance on external data or teacher models. RSIR operates in a closed loop: the current model generates plausible user interaction sequences, a fidelity-based quality control mechanism filters them for consistency with true user preferences, and a successor model is retrained on the enriched dataset. Our theoretical analysis shows that RSIR acts as a data-driven implicit regularizer, smoothing the optimization landscape and guiding models toward more robust solutions. Empirically, RSIR yields consistent, cumulative gains across multiple benchmarks and architectures. Notably, even smaller models benefit, and weak models can generate effective training curricula for stronger ones. These results demonstrate that recursive self-improvement is a general, model-agnostic approach to overcoming data sparsity, suggesting a scalable path forward for recommender systems and beyond. Our anonymized code is available at https://anonymous.4open.science/status/RSIR-7C5B.

## 1 Introduction

The paradigm of scaling models on ever-larger datasets is running into a bottleneck: the scarcity and cost of high-quality training data (Singh, 2023). This challenge spans domains from natural language processing (Dang et al., 2024) to computer vision (Wan et al., 2024), and it is especially acute in recommendation systems (Lai et al., 2024). Recommenders, which power modern digital platforms, must learn user preferences from interaction histories. Yet any given user engages with only a tiny fraction of a platform's catalog, leaving models with extremely sparse signals (Idrissi & Zellou, 2020). This sparsity produces rugged optimization landscapes, where models often converge to sharp, brittle minima that generalize poorly (Park & Tuzhilin, 2008; Gunathilaka et al., 2025).

A natural response is data augmentation. Prior work has enriched recommender training data through curated side information (e.g., metadata, reviews)(Cui et al., 2025) or by leveraging external "teach" models such as large language models(Luo et al., 2024). While effective in some cases, these approaches come with significant drawbacks: curated datasets are expensive and domain-specific, and reliance on massive teacher models introduces dependencies and risks of distributional mismatch with true user behavior. Another line of work explores heuristic augmentations such as item masking(Sun et al., 2019) or cropping(Xie et al., 2022), which provide only alternative views of existing data. Crucially, they do not generate novel, high-fidelity interaction sequences capable of densifying user trajectories.

This motivates a fundamentally different paradigm: **recursive self-improvement**. What if a model could use its own, partially learned understanding of user behavior to explore and generate its own training data? We propose to iteratively bootstrap a model's performance by leveraging its own predictive capabilities. The core idea is a synergistic loop: the current recommendation model is

Figure 1: Overview of the Recursive Self-Improving Recommendation (RSIR) Framework.

used to self-generate new plausible user histories, and a successor model is then retrained on this richer dataset. A stronger model generates better data, which in turn trains an even stronger model.

However, such a closed-loop system is inherently vulnerable to the amplification of its own biases and errors. An uncontrolled loop can quickly pollute the training set and lead to performance collapse(Shumailov et al., 2024; Alemohammad et al., 2024). To address this, we introduce a **fidelity-based quality control mechanism** that enforces bounded exploration: synthetic sequences must not only be novel but also remain faithful to a user's true interests. This prevents error amplification and ensures that the self-improvement process consistently produces useful data.

We instantiate this paradigm in the **Recursive Self-Improving Recommendation (RSIR)** framework. At each iteration, as shown in Fig. 1, RSIR (1) generates synthetic interaction sequences using the current model's predictive ability, (2) filters them via fidelity-based quality control, and (3) retrains a new model on the resulting high-quality dataset. Theoretically, we argue that RSIR functions as a data-driven implicit regularizer, smoothing the optimization landscape by reinforcing stable knowledge. Empirically, we show that RSIR improves performance across multiple benchmarks and architectures, including smaller models. Notably, even weaker models have the ability to bootstrap themselves, generating training data that can enhance the performance of stronger models. This underscores the efficiency and wide applicability of RSIR. Our contributions are as follows:

- We propose RSIR, the first framework that enables recommendation models to bootstrap their own training signals without reliance on external models or data.

- We introduce a mechanism that stabilizes recursive self-improvement by preventing error amplification and ensuring generated data remains faithful to user preferences.

- We provide a novel analysis showing that RSIR acts as an implicit regularizer that smooths the loss landscape, improving generalization.

- We conduct extensive experiments across diverse datasets and backbones, demonstrating that RSIR delivers consistent, cumulative performance gains and enables weak-to-strong transfer of synthetic training data.

## 2 RELATED WORKS

### 2.1 SELF-IMPROVING

Spurred by aspirations for general artificial intelligence, self-improvement has recently emerged as a major focus in machine learning. Building on this trend, both the areas of natural language processing and computer vision have adopted self-improvement strategies to develop generative models that can self-improve iteratively. For building self-improving LLMs, methods such as STaR (Zelikman et al., 2022), reinforced self-training (Gulcehre et al., 2023; Zhang et al., 2024), and self-rewarding (Yuan et al., 2024; Wang et al., 2024b) employ large language models to identify potential directions for self-improvement within their generated data, enabling the model to refine itself using its own outputs. This paradigm has also been extended beyond text. For example, RSIDiff (Zhang et al., 2025) applies self-generated data to recursively train diffusion models for state-of-the-art text-to-image generation, while STEP (Qiu et al., 2025) follows a similar self-improving paradigm to automatically produce reasoning-rich fine-tuning data from raw videos, thereby enhancing its own performance. Collectively, these developments exemplify a broader shift toward leveraging

models' internal mechanisms and outputs for continual self-improvement. However, most existing self-improving methods rely on evaluations beyond the generative model itself, such as large language models, external executors, and predefined rules, to assess data quality and iteratively refine their outputs. In contrast, our approach depends solely on the intrinsic characteristics of the dataset, achieving self-improvement by expanding decision boundaries. Moreover, to the best of our knowledge, we are the first to introduce a self-improving framework for data generation in the recommender systems domain.

## 2.2 SEQUENTIAL RECOMMENDATION

In recent years, recommender systems have attracted public attention and achieved substantial progress, generating considerable social and economic value, with the sequential recommendation system(SRS) being important due to its ability to leverage temporal dependencies in user–item interactions. Traditionally, SRS have been dominated by deep learning–based methods (Tang & Wang, 2018; Chang et al., 2021) which automatically learn rich representations and capture high-order interaction patterns for improved prediction of future behaviors. More recently, research has diversified into two directions: model-centric and data-centric approaches (Lai et al., 2024). Model-centric research increasingly focuses on generative architectures, particularly transformer-based decoders for modeling user interaction sequences (Zhai et al., 2024; Deng et al., 2025; Lee et al., 2025). Data-centric approaches, in contrast, emphasize improving the quality and utility of data itself and are often more effective than model-centric methods at alleviating sparsity and enhancing robustness. Within this paradigm, data augmentation techniques (Dang et al., 2025; Cui et al., 2025) introduce diverse perturbations or auxiliary signals into existing data in a heuristic manner to enhance model robustness and alleviate sparsity. Going beyond simple augmentation, data generation approaches (Liu et al., 2023; Yin et al., 2024; Lin et al., 2025) learn the underlying data distribution and leverage generative models to synthesize new interaction records, thereby enriching sparse datasets, improving model generalization, and better capturing complex user–item relationships. However, most current data-centric methods still depend on fixed external rules or one-shot processing and cannot sustain improvements in data quality over time. By contrast, our self-improving framework dispenses with external knowledge and, through iterative training, produces increasingly higher-quality data driven by the model's own understanding, forming a self-reinforcing loop.

## 3 METHODOLOGY

We formally introduce the Recursive Self-Improving Recommendation (RSIR) framework, a novel paradigm designed to mitigate data sparsity by enabling a model to iteratively refine its own training data. The central thesis is that a recommendation model, even one trained on sparse data, contains a nascent understanding of user preferences. RSIR operationalizes a feedback loop to cultivate this understanding, using the model itself to explore and generate plausible, high-fidelity user interaction sequences that densify the training landscape for its successor.

### 3.1 THE ITERATIVE SELF-IMPROVEMENT LOOP

Let $D_0 = \{s_u\}_{u \in U}$ be the initial training dataset, where $s_u = (i_1, i_2, \ldots, i_T)$ is the chronologically ordered interaction sequence for user $u$ from a global item set $I$. Our objective is to learn a sequence of increasingly powerful models, represented by their parameters $\theta_0, \theta_1, \ldots, \theta_K$, over $K$ iterations.

The RSIR process at iteration $k$ is defined by the following sequence:

1. **Model Training:** A recommendation model $f_{\theta_k}$ with parameters $\theta_k$ is trained on the current dataset $D_k$. For the initial iteration ($k = 0$), the model $f_{\theta_0}$ is trained on the original dataset $D_0$. The training objective is a standard next-item prediction task, maximizing the likelihood $P(i_t | s_{u,<t}; \theta_k)$.

2. **Synthetic Sequence Generation:** The trained model $f_{\theta_k}$ is employed as a generator to produce a set of synthetic user interaction sequences, $D'_{k+1}$. This generation process, detailed in Section 3.2, is the core of the self-improvement mechanism.

3. **Dataset Expansion:** The high-fidelity synthetic sequences are merged with the existing dataset to form an enriched training set for the next iteration: $D_{k+1} = D_k \cup D'_{k+1}$.

4. **Iterative Refinement:** A new model $f_{\theta_{k+1}}$ is initialized and trained from scratch on the augmented dataset $D_{k+1}$.

This recursive loop can be expressed as:

$$\theta_k \xrightarrow{\text{Generate}} D'_{k+1} \xrightarrow{\text{Expand}} D_{k+1} \xrightarrow{\text{Train}} \theta_{k+1}$$

systematically producing a trajectory of models $(\theta_0, \theta_1, \ldots, \theta_K)$, with each trained on an increasingly rich, broader data distribution. The pseudo code is shown in the Appendix A.

## 3.2 PRINCIPLED SYNTHETIC SEQUENCE GENERATION

The efficacy of RSIR hinges on the ability to generate sequences that are not only novel but also faithful to plausible user behavior. Generating random, unconstrained sequences would quickly introduce noise and lead to catastrophic performance collapse. To avoid this, we propose a generation process built on two principles: **bounded exploration** and **fidelity-based quality control**.

For each user sequence $s_u \in D_k$, we generate $m$ synthetic trajectories by autoregressively extending an initial context. The process begins by seeding the generation with a prefix of the user's true history, $S_{ctx} = (i_1, \ldots, i_j)$, where $j$ is chosen randomly.

### 3.2.1 BOUNDED EXPLORATION VIA A HYBRID CANDIDATE POOL

At each generation step $t$, the model $f_{\theta_k}$ predicts a probability distribution over the next item given the current context $S_{ctx}$. To balance the discovery of new patterns with adherence to established preferences, we perform top-k sampling from a hybrid candidate pool constructed as follows:

> **Bounded Exploration**
>
> - **Exploitation:** With probability $p$, candidates are sampled from the user's historical interactions $s_u$. This encourages the model to find novel sequential patterns and higher-order connections within items the user has already engaged with.
>
> - **Exploration:** With probability $1-p$, candidates are sampled from the global item set $I$. This allows the model to extrapolate beyond the user's known interactions, cautiously expanding the boundaries of their preference profile.

This hybrid strategy facilitates a form of **bounded exploration**, preventing the model from generating entirely random sequences while still allowing for the discovery of novel, plausible interests.

### 3.2.2 FIDELITY-BASED QUALITY CONTROL

To prevent the iterative loop from amplifying model biases and drifting into implausible regions of the data space, we introduce a critical safeguard. After sampling a candidate item $i_{gen,t}$, we provisionally update the context to $S'_{ctx} = S_{ctx} \cup \{i_{gen,t}\}$. We then verify if this synthetic step remains consistent with the user's true future interests.

Formally, let $S_{tgt} = s_u \setminus S_{ctx}$ be the set of ground-truth future items in the original sequence. We accept the generated item $i_{gen,t}$ if and only if at least one true future item is still ranked highly by the model, given the new synthetic context:

$$\boxed{\exists i_j \in S_{tgt} \quad \text{such that} \quad \text{Rank}_{f_{\theta_k}}(i_j | S'_{ctx}) \leq \tau} \tag{1}$$

where $\text{Rank}_{f_{\theta_k}}(i_j | S'_{ctx})$ is the predicted rank of item $i_j$ by model $f_{\theta_k}$ given the context $S'_{ctx}$, and $\tau$ is a hyperparameter defining the rank threshold.

If this condition is satisfied, the step is deemed high-fidelity. The item $i_{gen,t}$ is appended to the synthetic sequence, and the context is updated ($S_{ctx} \leftarrow S'_{ctx}$) for the next generation step. If the condition fails, it signals that the generated sequence is beginning to diverge from the user's underlying preferences. The generation for this specific sequence is immediately terminated to prevent low-quality data from polluting the training set.

This mechanism acts as a crucial regularizer, ensuring that the self-generated data remains "on-manifold" with respect to the user's true dynamics, thereby stabilizing the self-improvement loop and guaranteeing the integrity of the augmented dataset. Finally, all successfully generated sequences are collected to form $D'_{k+1}$, after filtering for duplicates and minimum length requirements.

## 3.3 COMPUTATIONAL COMPLEXITY ANALYSIS

We analyze the time complexity of RSIR over $K$ iterations. Let $N$ be the number of user sequences, $L$ the maximum sequence length, $d$ the hidden dimension, and $|\mathcal{V}|$ the item vocabulary size. The computational cost for the backbone model to process one sequence is $\mathcal{C}_{\text{model}} \approx O(L^2 d + L d^2)$.

The total complexity consists of two phases: **Model Training** and **Sequence Generation**. For training, since the dataset size $N_k$ grows iteratively, the cumulative complexity is $\sum_{k=0}^{K} O(N_k \cdot \mathcal{C}_{\text{model}})$. For generation, performing $m$ attempts per sequence incurs a cost dominated by the fidelity check. While the theoretical worst-case is $O(d|\mathcal{V}|)$ per step, our "Break" mechanism (Sec. 3.2.2) acts as an adaptive pruner, restricting the effective generated length to $L_e \ll L$. Additionally, for large vocabularies, the linear scan can be optimized to $O(d \log |\mathcal{V}|)$ via approximate retrieval.

Consequently, the overall complexity of RSIR is $O\left(N_k \cdot (E \cdot \mathcal{C}_{\text{model}} + m \cdot L_e \cdot (\mathcal{C}_{\text{model}} + d|\mathcal{V}|))\right)$, which is strictly bounded. The total runtime scales linearly with respect to vocabulary size $V$ and generation length $L_e$, ensuring practical feasibility and scalability. A rigorous derivation and empirical runtime analysis are provided in Appendix D.

## 4 DISCUSSION AND THEORETICAL ANALYSIS

In this section, we provide a theoretical grounding for our Recursive Self-Improving Recommendation (RSIR) framework. A primary challenge hindering recommendation systems is extreme data sparsity, which forces models to learn from a fragmented signal, often leading them to overfit on spurious correlations and converge in sharp, brittle minima of the loss landscape. Our RSIR framework directly addresses this by enabling the model to perform a form of **bounded exploration**. It explores the boundaries of its own knowledge by generating novel interaction sequences, but this exploration is constrained by our fidelity-based quality control (Sec. 3.2.2). This mechanism ensures the exploration is reliable and faithful to the user's underlying interests, effectively and safely densifying the data space around known user trajectories.

### 4.1 IMPLICIT REGULARIZATION AND LANDSCAPE SMOOTHING

This generation strategy directly impacts the optimization dynamics. The fidelity-based quality control acts as a filter for model stability; a model with parameters $\theta_k$ in a sharp minimum would fail the check, as its representations are too brittle to handle contextual perturbations. Therefore, a synthetic sequence $s'$ is included in the generated set $D'_{k+1}$ only if the model $f_{\theta_k}$ is robust in its vicinity. This implies that the aggregate loss on the generated set,

$$L_{\text{gen}}(\theta) = \frac{1}{|D'_{k+1}|} \sum_{s' \in D'_{k+1}} l\big(f_\theta(s')\big), \tag{2}$$

defines a loss surface that is exceptionally smooth and low-curvature around the current solution $\theta_k$.

The iterative refinement step then optimizes a composite objective:

$$\theta_{k+1} = \arg\min_\theta \left[L_k(\theta) + \lambda\, L_{\text{gen}}(\theta)\right], \tag{3}$$

where $L_k(\theta)$ is the loss on the existing (sparse) data from $D_k$. The $L_{\text{gen}}(\theta)$ term, derived from the densified data, is approximately equivalent to a regularized optimization on the original landscape:

$$\arg\min_\theta \left[L_k(\theta) + \Omega\big(\theta; \theta_k\big)\right], \tag{4}$$

Here, $\Omega(\theta; \theta_k)$ is an **implicit regularizer** that penalizes sharpness (i.e., high curvature) by encouraging the model to reach flatter minima. In Appendix E.1, we formally prove that this regularizer operates geometrically as a **Manifold Tangential Gradient Penalty**, minimizing the gradient norm specifically along the directions of the user preference manifold, rather than blindly suppressing all parameter updates. This leads to our first key insight.

> **Insight 1: RSIR as an Implicit Regularizer**
>
> **RSIR functions as a data-driven implicit regularizer. It smooths the loss landscape by forcing the optimizer to find wider, flatter minima aligned with the user preference manifold that generalize better.**

## 4.2 ERROR ANALYSIS AND STABILITY GUARANTEE

Beyond the geometric interpretation, a critical question remains: does training on self-generated data lead to error accumulation? In Appendix E.2, we derive the recursive error bound for the RSIR framework. We prove that the generalization error $\mathcal{E}(\theta_{k+1})$ is bounded by a linear contraction of the previous error $\mathcal{E}(\theta_k)$, subject to a noise term introduced by potential hallucinations.

$$\mathcal{E}(\theta_{k+1}) \leq (1-\lambda)\mathcal{E}_0 + \lambda \left[ \underbrace{(1-\tilde{p}_k)\rho\mathcal{E}(\theta_k)}_{\text{Contraction from Valid Exploration}} + \underbrace{\tilde{p}_k\mathcal{E}_{\max}}_{\text{Leakage Penalty}} \right] \tag{5}$$

Crucially, we identify a **Breakdown Point** for the fidelity leakage rate $\tilde{p}_k$. Convergence is guaranteed if and only if the fidelity check is strict enough to keep noise below this threshold. **Furthermore, the analysis reveals an "irreducible noise floor" due to non-zero leakage $\tilde{p}_k$.** As the model improves ($\mathcal{E}(\theta_k) \to 0$), the marginal benefit of contraction diminishes while the noise penalty persists. This explains why performance may plateau or slightly degrade in late-stage iterations if the noise floor outweighs the shrinking gain, underscoring the necessity of our strict fidelity control.

This analysis reframes RSIR from simple data augmentation to a sophisticated, model-guided regularization strategy. Instead of relying on external knowledge from a powerful teacher model, RSIR demonstrates that a model can bootstrap its own performance by generating its own curriculum. This directly informs our central thesis about the nature of self-improvement.

> **Insight 2: Self-Improvement is Not Just for Large Models**
>
> **Effective self-improvement is not an emergent capability of large models, but a fundamental benefit of recursive regularization that is accessible to any model architecture.**

## 5 EXPERIMENTS

### 5.1 EXPERIMENTAL SETTINGS

#### 5.1.1 DATASETS

We evaluate our framework on four public benchmark datasets: **Beauty**, **Sports**, and **Toys** from the Amazon review dataset[1], and **Yelp**[2]. These datasets are widely used as standard benchmarks for sequential recommendation tasks (Yin et al., 2024; Xie et al., 2024; Kim et al., 2025). They are primarily characterized by high data sparsity, which makes them an ideal testbed for evaluating RSIR's ability to address this core challenge. Dataset statistics are provided in Appendix B.3.

#### 5.1.2 BACKBONES AND BASELINE MODELS

To demonstrate the broad applicability of RSIR, we integrate it with three representative sequential recommendation models. The backbone models are as follows: the Transformer-based model SAS-Rec(Kang & McAuley, 2018), the Contrastive Learning-based model CL4SRec(Xie et al., 2022), and the Generative Model-based model HSTU(Zhai et al., 2024). For a detailed description of the method, please refer to Appendix B.1.

Our primary evaluation focuses on the performance gains achieved when applying RSIR to these backbones. As our work introduces the first recursive self-improvement paradigm, we compare

---

[1]http://jmcauley.ucsd.edu/data/amazon/
[2]https://www.yelp.com/dataset

Table 1: Performance Comparison on Three Backbone Models. The Best and Second-best Results Are Shown in Bold and Underlined. **RSIR-FT** and **RSIR** denote the fine-tuning variant and the re-training version of our method, respectively. The 'Improv' row reports the relative improvement of our methods (**RSIR-FT** or **RSIR**) compared to the best baseline. (p-value $< 0.05$)

| | Method | amazon-toys | | amazon-beauty | | amazon-sport | | yelp | |
|---|---|---|---|---|---|---|---|---|---|
| | | NDCG@10 | Recall@10 | NDCG@10 | Recall@10 | NDCG@10 | Recall@10 | NDCG@10 | Recall@10 |
| SASRec | Base | 0.0477 | 0.0795 | 0.0290 | 0.0548 | 0.0271 | 0.0474 | 0.0183 | 0.0371 |
| | +Reordering | 0.0488 | 0.0831 | 0.0285 | 0.0520 | 0.0265 | 0.0465 | 0.0186 | 0.0373 |
| | +Insertion | 0.0493 | 0.0834 | 0.0295 | 0.0545 | 0.0276 | 0.0472 | 0.0190 | 0.0379 |
| | +ASReP | 0.0492 | 0.0820 | 0.0286 | 0.0522 | 0.0282 | 0.0481 | 0.0188 | 0.0373 |
| | +DiffuASR | 0.0480 | 0.0806 | 0.0298 | 0.0554 | 0.0279 | 0.0475 | 0.0186 | 0.0366 |
| | +DR4SR | 0.0499 | 0.0830 | 0.0300 | 0.0557 | 0.0286 | 0.0495 | 0.0191 | 0.0378 |
| | +RSIR-FT | 0.0507 | 0.0860 | **0.0322** | **0.0594** | 0.0290 | 0.0500 | **0.0200** | 0.0393 |
| | +RSIR | **0.0508** | **0.0872** | 0.0303 | 0.0578 | **0.0293** | **0.0512** | **0.0200** | **0.0399** |
| | Improv | 1.80% | 4.56% | 7.33% | 6.64% | 2.45% | 3.43% | 4.71% | 5.28% |
| CL4SRec | Base | 0.0519 | 0.0870 | 0.0307 | 0.0579 | 0.0284 | 0.0491 | 0.0205 | 0.0392 |
| | +Reordering | 0.0514 | 0.0868 | 0.0303 | 0.0565 | 0.0283 | 0.0488 | 0.0208 | 0.0407 |
| | +Insertion | 0.0532 | 0.0877 | 0.0294 | 0.0550 | 0.0288 | 0.0495 | 0.0200 | 0.0397 |
| | +ASReP | 0.0518 | 0.0873 | 0.0306 | 0.0575 | 0.0289 | 0.0481 | 0.0198 | 0.0388 |
| | +DiffuASR | 0.0482 | 0.0808 | 0.0308 | 0.0582 | 0.0288 | 0.0487 | 0.0198 | 0.0392 |
| | +DR4SR | 0.0535 | 0.0887 | 0.0310 | 0.0590 | 0.0289 | 0.0500 | 0.0213 | 0.0416 |
| | +RSIR-FT | 0.0541 | 0.0926 | **0.0344** | **0.0649** | **0.0301** | **0.0523** | 0.0219 | 0.0422 |
| | +RSIR | **0.0543** | **0.0927** | 0.0318 | 0.0596 | 0.0297 | 0.0517 | **0.0224** | **0.0441** |
| | Improv | 1.50% | 4.51% | 10.97% | 10.00% | 4.15% | 4.60% | 5.16% | 6.01% |
| HSTU | Base | 0.0512 | 0.0869 | 0.0302 | 0.0568 | 0.0285 | 0.0492 | 0.0192 | 0.0373 |
| | +Reordering | 0.0497 | 0.0837 | 0.0308 | 0.0558 | 0.0282 | 0.0482 | 0.0198 | 0.0384 |
| | +Insertion | 0.0501 | 0.0871 | 0.0302 | 0.0563 | 0.0284 | 0.0493 | 0.0197 | 0.0386 |
| | +ASReP | 0.0487 | 0.0815 | 0.0288 | 0.0537 | 0.0284 | 0.0483 | 0.0195 | 0.0379 |
| | +DiffuASR | 0.0462 | 0.0785 | 0.0310 | 0.0578 | 0.0288 | 0.0497 | 0.0192 | 0.0379 |
| | +DR4SR | 0.0507 | 0.0867 | 0.0304 | 0.0567 | 0.0294 | 0.0515 | 0.0196 | 0.0384 |
| | +RSIR-FT | 0.0536 | 0.0914 | **0.0324** | **0.0599** | 0.0299 | 0.0521 | 0.0204 | 0.0403 |
| | +RSIR | **0.0544** | **0.0924** | **0.0324** | 0.0596 | **0.0305** | **0.0531** | **0.0209** | **0.0411** |
| | Improv | 6.25% | 6.08% | 4.52% | 3.63% | 3.74% | 3.11% | 5.56% | 6.48% |

against two common heuristic-based data augmentation methods and three learnable data generation methods, which represent the closest alternative for enriching the training data without external models or knowledge. For a detailed description of the method, please refer to Appendix B.2.

### 5.1.3 IMPLEMENTATION DETAILS

We adopt the leave-one-out strategy for evaluation (last item for test, second-to-last for validation). For evaluating retrieval performance, we use NDCG@K, Recall@K as metrics, which are widely used in related works (He et al., 2017; 2020), and we set the K value to 10 and 20. We train for a maximum of 1000 epochs with an early stopping patience of 20. All models are implemented using the RecStudio framework (Lian et al., 2023) and trained on a single GPU. For the RSIR process, we employ a grid search to find the optimal hyperparameters for the fidelity threshold $\tau \in \{1, 3, 5, 10, 20, 50, 100\}$, the number of generation attempts per sequence $m \in \{5, 10, 20\}$, and the historical sampling probability $p \in \{0.0, 0.2, 0.4, 0.5, 0.6, 0.8, 1.0\}$. The general paradigm for sequential recommendation and details of our experimental setup are presented in Appendix H.

### 5.2 MAIN RESULTS: EFFICACY OF RSIR

### 5.2.1 SINGLE-ITERATION PERFORMANCE

First, we investigate the core premise of our work: whether a model can effectively improve itself by training on its own generated data. As shown in Table 1, applying a single iteration of RSIR yields consistent and significant performance improvements across all three backbone models and all four datasets. For instance, RSIR improves the Recall@10 of the powerful HSTU model by 7.71% on Sports and 7.14% on Yelp. This result empirically confirms our central hypothesis from Section 4. RSIR's bounded exploration generates high-fidelity data that densifies meaningful user trajectories, which in turn enables the model to find a more generalizable solution. Furthermore,

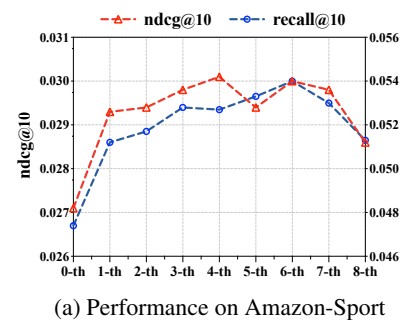
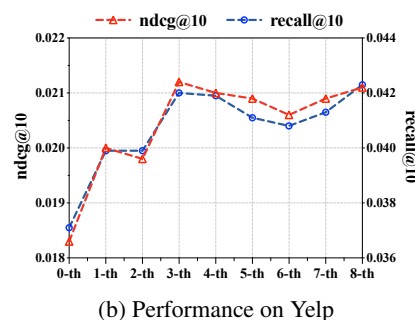

(a) Performance on Amazon-Sport      (b) Performance on Yelp

Figure 2: Performance of RSI Across Different Iterations on Amazon-Sport and Yelp.

RSIR consistently outperforms the heuristic-based data augmentation baselines. This demonstrates that principled, model-guided generation is superior to simply increasing data volume with noisy or uninformative sequences (e.g., via item insertion or reordering).

> **Result 1.** *RSIR provides significant, model-agnostic performance gains in a single iteration.*

### 5.2.2 RECURSIVE MULTI-ITERATION PERFORMANCE

We further explore if these gains compound over multiple iterations. Fig. 2 plots model performance over the RSIR recursion. The results clearly show that performance continues to rise through several cycles. On the Sports dataset, the initial 8.02% gain in Recall@10 for HSTU extends to 13.92% after three iterations. This powerfully demonstrates the virtuous cycle of the recursive loop: **a stronger model generates higher-quality data, which in turn trains an even stronger successor.** Performance eventually saturates, which we attribute to the gradual amplification of systemic model biases outweighing the benefits of data densification. Despite this, the substantial multi-iteration gains affirm the efficacy and power of the recursive process.

> **Result 2.** *RSIR's gains are cumulative across multiple iterations, validating the core recursive mechanism where model improvement and data quality mutually reinforce each other.*

## 5.3 ABLATION AND ANALYSIS

### 5.3.1 THE CRITICAL ROLE OF FIDELITY-BASED QUALITY CONTROL.

To verify the importance of our fidelity-based quality control module, we conduct an ablation study where it is removed (i.e., all generated items are accepted). As shown in Table 2, while uncontrolled generation shows marginal gains in the first iteration, it leads to catastrophic performance collapse in subsequent iterations. This is because model errors and biases are amplified without constraint, rapidly polluting the training data. This result validates that **bounded exploration is critical**; simply increasing data volume with unconstrained generation is harmful.

Table 2: Ablation results on *amazon-sport*. 'w/o' denotes without the fidelity-based quality control module. (p-value $< 0.05$)

|  |  | NDCG@10 | Recall@10 |
|---|---|---|---|
| SASRec |  | 0.0271 | 0.0474 |
| RSIR-1th | w/o | 0.0273 | 0.0472 |
|  | w | 0.0293 | 0.0512 |
| RSIR-2th | w/o | 0.0209 | 0.0384 |
|  | w | 0.0294 | 0.0517 |
| RSIR-3th | w/o | 0.0119 | 0.0210 |
|  | w | 0.0298 | 0.0528 |

Furthermore, Fig. 3a analyzes the sensitivity to the fidelity threshold $\tau$. The general performance trend illustrates the crucial trade-off between generation diversity and data fidelity. Overly strict thresholds ($\tau \to 1$) choke the model, preventing it from generating diverse sequences, while overly permissive thresholds ($\tau \to \infty$) allow noisy, low-fidelity data into the training set, both of which degrade performance.

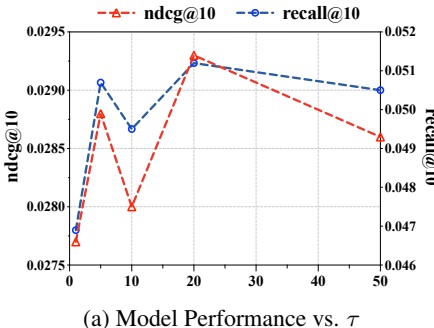 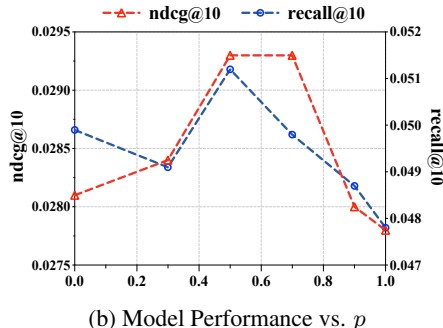

(a) Model Performance vs. $\tau$           (b) Model Performance vs. $p$

Figure 3: Comparison of Model Performance with Respect to Different Parameters.

### 5.3.2 ANALYSIS OF THE BOUNDED EXPLORATION STRATEGY

Fig. 3b shows the impact of the historical sampling probability $p$, which governs the exploitation-exploration trade-off. Performance peaks around $p = 0.5$. Pure exploitation ($p = 1.0$) fails to expand the model's knowledge boundary by discovering novel interests, while pure exploration ($p = 0.0$) is inefficient and risks generating irrelevant data that would be filtered by the quality control. This confirms that the most effective data is generated when the model is encouraged to both find new connections within known interests and cautiously explore beyond them.

> **Result 3.** *Principled data generation, governed by strict fidelity control and a balanced exploration strategy, is essential for stable and effective self-improvement.*

### 5.4 CAN WEAKER MODELS TEACH STRONGER MODELS?

First, we validate the core premise of our recursive framework: a model's ability to generate high-quality data improves as it becomes stronger. Observing the rows of the heatmap, we see a clear trend: for any given student, a stronger teacher model provides a superior training curriculum. This empirically confirms the logic behind our recursive loop—the pursuit of iterative self-improvement is the optimal path to maximizing absolute performance.

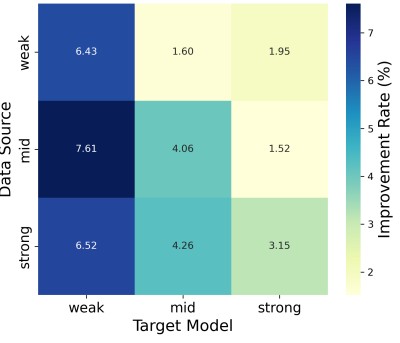

Figure 4: Improvement Rate Heatmap.

Second, and more strikingly, the process itself is fundamentally effective, regardless of the teacher's capacity. The results show that even a weak teacher provides a significant +1.95% performance lift to a strong student. This is a crucial finding that directly confirms our theoretical conclusion from Sec. 4: the primary benefit of RSIR stems from the process of recursive regularization itself. The targeted data densification and landscape smoothing are effective even when the generating model has limited power.

These two findings offer a powerful dual perspective on RSIR. The first finding justifies the recursive loop as the best strategy for achieving state-of-the-art performance. The second highlights the framework's notable potential for practice, where a computationally inexpensive model can be used to generate a powerful training curriculum for a large-scale production model, balancing performance gains with resource constraints.

> **Result 4.** *Stronger models are better teachers, yet even weak models can significantly improve stronger ones.*

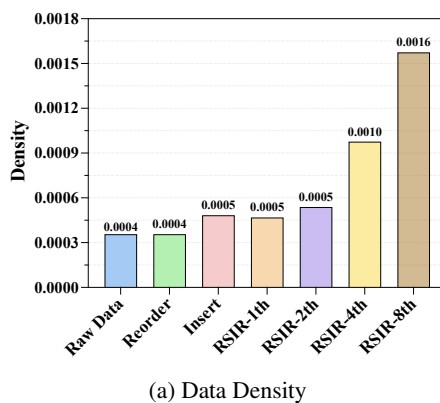 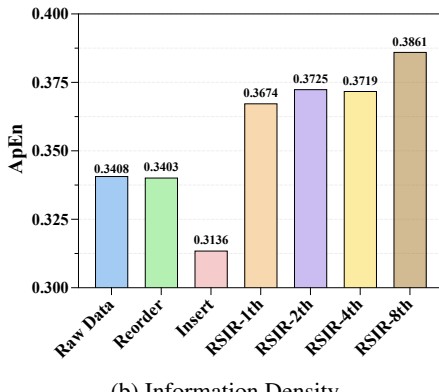

(a) Data Density  (b) Information Density

Figure 5: Generated Data Analysis.

## 5.5 ANALYSIS OF GENERATED DATA

To provide direct, data-level evidence for RSIR's efficacy, we analyze the properties of the generated sequences. First, we confirm that RSIR directly addresses the problem of **data sparsity**. As shown in Fig. 5a, the density of the training data increases progressively with each RSIR iteration, reaching a +342.14% improvement after eight iterations.

However, merely increasing data density is insufficient, as this may introduce noise and degrade performance. To measure the quality and informativeness of the generated data, we employ Approximate Entropy (ApEn) (shown in Appendix G)(Pincus, 1991; Shen et al., 2024), a metric for sequence complexity. As shown in Fig. 5b, RSIR consistently increases the ApEn of the dataset, demonstrating that the newly generated sequences are rich in information and add novel patterns.

This stands in stark contrast to the heuristic "Insertion" baseline. While Insertion also increases data density, it simultaneously decreases the dataset's ApEn. This provides quantitative proof that naive augmentation pollutes the training set with simple, uninformative noise. RSIR, on the other hand, generates not just more data, but fundamentally better data.

> **Result 5.** *RSIR addresses data sparsity in a principled manner by generating sequences that are both voluminous and information-rich.*

## 6 CONCLUSION

In this work, we tackled the fundamental challenge of extreme data sparsity in recommendation systems. We proposed the Recursive Self-Improving Recommendation (RSIR) framework, which enables models to bootstrap their own performance by iteratively generating and refining training data without reliance on external sources. A fidelity-based quality control mechanism stabilizes this loop, ensuring that synthetic interactions remain faithful to user preferences and preventing error amplification. Our theoretical analysis shows that RSIR functions as a data-driven implicit regularizer, smoothing the optimization landscape and guiding models toward robust solutions. Experiments across multiple benchmarks and architectures confirm that RSIR delivers consistent, cumulative gains, with fidelity control playing a critical role. Notably, even weak models can generate effective training curricula for stronger models.

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

## A  PSEUDO CODE FOR RSIR FRAMEWORK

---

**Algorithm 1:** Recursive Self-Improving Recommendation Framework

---

**Input:** $D_0$: initial dataset; $f_\theta$: recommendation model; $m$: number of synthetic sequences per user; $T$: maximum sequence length; $p$: Exploitation probability; $K$: number of iterations; $\tau$: rank threshold.

**Output:** Final augmented dataset $D_K$

**for** $k = 0, 1, 2, \ldots, K-1$ **do**
  // Phase 1:  Model Training
  Train model $f_{\theta_k}$ on $D_k$:
  // Phase 2:  Quality Control Generation
  **for** *user sequence* $s_u = (i_1, \ldots, i_T)$ *in* $D_k$ **do**
    **for** $j = 1$ *to* $m$ **do**
      // $S_{ctx}$:  current context
      // $S_{tgt}$:  remaining true items
      Initialize $S_{ctx} \leftarrow (i_1)$, $S_{tgt} \leftarrow s_u$;
      **for** $t = 2$ *to* $T$ **do**
        Construct hybrid candidate pool $\mathcal{C}$:
          **Exploitation with prob.** $p$ : sample from user's history
          **Exploration with prob.** $1 - p$ : sample from global item set
        Generate next item $i_{gen,t} \sim f_{\theta_k}(S_{ctx})$ from $\mathcal{C}$;
        Form new context $S'_{ctx} \leftarrow S_{ctx} \cup \{i_{gen,t}\}$;
        **if** $\exists i_j \in S_{tgt}$ *such that* $\mathrm{Rank}_{f_{\theta_k}}(i_j | S'_{ctx}) \leq \tau$ **then**
          Update $S_{ctx} \leftarrow S'_{ctx}$;
          Update $S_{tgt} \leftarrow S_{tgt} \setminus \{i_{gen,t}\}$;
        **else**
          Break;
      **if** $|S_{ctx}| \geq 2$ *and* $S_{ctx}$ *not duplicate* **then**
        add $S_{ctx}$ to $D'_{k+1}$;
  // Phase 3:  Data Expansion
  Form new training set $D_{k+1} \leftarrow D_k \cup D'_{k+1}$;

---

## B  BASELINES AND BENCHMARK DATASETS STATISTICS

### B.1  BACKBONES

The three baselines we used are described as follows:

- **SASRec**(Kang & McAuley, 2018): a widely adopted Transformer-based model for sequential recommendation, which leverages self-attention to capture user interaction patterns.

- **CL4SRec**(Xie et al., 2022): a contrastive learning–enhanced sequential recommendation model that augments user interaction sequences to improve representation learning.

- **HSTU**(Zhai et al., 2024): a SOTA generative recommendation model that employs hierarchical self-attention to efficiently model long and heterogeneous user interaction sequences.

## B.2 BASELINES

- **Heuristic-based Data Augmentation**
    - **Reordering**(Zhou et al., 2024): Randomly shuffles items within a subsequence.
    - **Insertion**(Liu et al., 2021a): Adds items to the original sequence.
- **Learnable Data Generation**
    - **ASREP**(Liu et al., 2021b): Extends sequence length via forward generation.
    - **DiffuASR**(Liu et al., 2023): Diffusion-based data generation.
    - **DR4SR**(Yin et al., 2024): Augments data quantity by regenerating new sequences.

## B.3 DATASET STATISTIC

Table 3 showcases the statistics of four benchmark datasets after 5-core filtering. Avg. length indicates the average number of interactions per user.

Table 3: Statistics of Benchmark Datasets after Preprocessing.

| Dataset | amazon-toys | amazon-beauty | amazon-sport | yelp |
|---|---|---|---|---|
| $U$ | 19,412 | 22,363 | 35,598 | 30,431 |
| $V$ | 11,876 | 12,066 | 18,281 | 20,014 |
| # Interactions | 106,254 | 127,598 | 187,694 | 216,733 |
| Avg. length | 5.47 | 5.71 | 5.27 | 7.12 |
| Sparsity | 0.999539 | 0.999527 | 0.999712 | 0.999644 |

## C DETAILED EXPERIMENT RESULTS

### C.1 GENERATED DATASET STATISTICS

Table 4 shows the scale and sparsity of the expanded datasets, generated after one iteration of self-improvement for different backbone generative models on four datasets, and compares them with the scale and sparsity of the original datasets.

Table 4: Dataset Statistics: Original vs. Generated via Different Backbone Models.

| Dataset | amazon-toys | | | | amazon-beauty | | | |
|---|---|---|---|---|---|---|---|---|
| | Original | SASRec | CL4SRec | HSTU | Original | SASRec | CL4SRec | HSTU |
| Sequences | 19412 | 21728 | 23942 | 27244 | 22363 | 32684 | 35162 | 28351 |
| U | 19412 | 19412 | 19412 | 19412 | 22363 | 22363 | 22363 | 22363 |
| V | 11876 | 11876 | 11876 | 11876 | 12066 | 12066 | 12066 | 12066 |
| Interactions | 106254 | 112582 | 121512 | 130880 | 127598 | 178051 | 185255 | 151179 |
| Sparsity | 0.999539 | 0.999512 | 0.999473 | 0.999432 | 0.999527 | 0.999340 | 0.999313 | 0.999440 |

| Dataset | amazon-sport | | | | yelp | | | |
|---|---|---|---|---|---|---|---|---|
| | Original | SASRec | CL4SRec | HSTU | Original | SASRec | CL4SRec | HSTU |
| Sequences | 35598 | 50233 | 51291 | 52357 | 30431 | 47810 | 48250 | 33868 |
| U | 35598 | 35598 | 35598 | 35598 | 30431 | 30431 | 30431 | 30431 |
| V | 18281 | 18281 | 18281 | 18281 | 20014 | 20014 | 20014 | 20014 |
| Interactions | 187694 | 239636 | 243094 | 246004 | 216733 | 285178 | 289004 | 225716 |
| Sparsity | 0.999712 | 0.999632 | 0.999626 | 0.999622 | 0.999644 | 0.999532 | 0.999525 | 0.999629 |

### C.2 PERFORMANCE COMPARISON WITH DATA-CENTRIC METHODS

We evaluated our method against traditional data augmentation on four datasets using different backbone models. Table 5 shows the results over four datasets, measured by NDCG@20 and Recall@20. The 'Improv' row indicates the relative improvement of our method over the augmentation baselines. It is important to note that our method was run for only a single self-improvement iteration.

Table 5: Performance Comparison on Three Backbone Models (Metrics @20). The Best and Second-best Results Are Shown in Bold and Underlined. **RSIR-FT** and **RSIR** denote the fine-tuning variant and the re-training version of our method, respectively. The 'Improv' row reports the relative improvement of our methods compared to the best baseline. (p-value $< 0.05$)

| | Method | amazon-toys | | amazon-beauty | | amazon-sport | | yelp | |
|---|---|---|---|---|---|---|---|---|---|
| | | NDCG@20 | Recall@20 | NDCG@20 | Recall@20 | NDCG@20 | Recall@20 | NDCG@20 | Recall@20 |
| SASRec | Base | 0.0553 | 0.1095 | 0.0359 | 0.0821 | 0.0320 | 0.0669 | 0.0240 | 0.0599 |
| | +Reordering | 0.0551 | 0.1084 | 0.0343 | 0.0751 | 0.0314 | 0.0661 | 0.0248 | 0.0619 |
| | +Insertion | 0.0560 | 0.1102 | 0.0361 | 0.0806 | 0.0327 | 0.0672 | 0.0246 | 0.0603 |
| | +ASReP | 0.0559 | 0.1089 | 0.0350 | 0.0779 | 0.0332 | 0.0681 | 0.0248 | 0.0609 |
| | +DiffuASR | 0.0545 | 0.1064 | 0.0364 | 0.0814 | 0.0328 | 0.0669 | 0.0244 | 0.0599 |
| | +DR4SR | 0.0564 | 0.1106 | 0.0367 | 0.0816 | 0.0337 | 0.0696 | 0.0246 | 0.0599 |
| | +RSIR-FT | **0.0578** | **0.1135** | **0.0394** | **0.0879** | 0.0342 | 0.0708 | **0.0261** | **0.0637** |
| | +RSIR | 0.0573 | 0.1133 | 0.0373 | 0.0858 | **0.0345** | **0.0717** | 0.0259 | **0.0637** |
| | Improv | 2.48% | 2.62% | 7.36% | 7.06% | 2.37% | 3.02% | 5.24% | 2.91% |
| CL4SRec | Base | 0.0599 | 0.1186 | 0.0378 | 0.0862 | 0.0331 | 0.0679 | 0.0267 | 0.0639 |
| | +Reordering | 0.0582 | 0.1139 | 0.0368 | 0.0824 | 0.0331 | 0.0679 | 0.0272 | 0.0662 |
| | +Insertion | 0.0610 | 0.1187 | 0.0363 | 0.0822 | 0.0335 | 0.0684 | 0.0259 | 0.0631 |
| | +ASReP | 0.0587 | 0.1144 | 0.0374 | 0.0843 | 0.0337 | 0.0674 | 0.0258 | 0.0628 |
| | +DiffuASR | 0.0547 | 0.1066 | 0.0384 | 0.0881 | 0.0340 | 0.0693 | 0.0256 | 0.0625 |
| | +DR4SR | 0.0610 | 0.1184 | 0.0386 | 0.0880 | 0.0337 | 0.0694 | 0.0276 | 0.0666 |
| | +RSIR-FT | **0.0615** | **0.1223** | **0.0440** | **0.0961** | **0.0353** | 0.0730 | 0.0282 | 0.0674 |
| | +RSIR | 0.0613 | 0.1222 | 0.0392 | 0.0890 | 0.0352 | **0.0734** | **0.0288** | **0.0693** |
| | Improv | 0.82% | 3.03% | 13.99% | 9.08% | 3.82% | 5.76% | 4.35% | 4.05% |
| HSTU | Base | 0.0580 | 0.1135 | 0.0370 | 0.0838 | 0.0338 | 0.0704 | 0.0250 | 0.0602 |
| | +Reordering | 0.0570 | 0.1125 | 0.0371 | 0.0811 | 0.0329 | 0.0671 | 0.0256 | 0.0616 |
| | +Insertion | 0.0573 | 0.1154 | 0.0377 | 0.0862 | 0.0336 | 0.0701 | 0.0252 | 0.0606 |
| | +ASReP | 0.0555 | 0.1086 | 0.0349 | 0.0780 | 0.0338 | 0.0697 | 0.0254 | 0.0614 |
| | +DiffuASR | 0.0529 | 0.1052 | 0.0379 | 0.0845 | 0.0342 | 0.0697 | 0.0252 | 0.0616 |
| | +DR4SR | 0.0576 | 0.1136 | 0.0377 | 0.0840 | 0.0346 | 0.0726 | 0.0253 | 0.0611 |
| | +RSIR-FT | 0.0608 | 0.1199 | **0.0394** | **0.0878** | 0.0358 | 0.0745 | 0.0268 | 0.0640 |
| | +RSIR | **0.0620** | **0.1223** | 0.0389 | 0.0851 | **0.0363** | **0.0762** | **0.0272** | **0.0660** |
| | Improv | 6.90% | 5.98% | 3.96% | 1.86% | 4.91% | 4.96% | 6.25% | 7.14% |

## C.3 EVALUATION ON COMPREHENSIVE METRICS

In the main text, we primarily adopted NDCG and Recall as evaluation metrics, following standard conventions in sequential recommendation. However, to provide a more holistic view of the model's performance and ensure that the improvements are robust across different evaluation perspectives, we extend our analysis to include **Precision**, **F1-score**, and **Mean Reciprocal Rank (MRR)**.

Table 6 presents the performance comparison between the Base model (SASRec) and the RSIR-enhanced model across four datasets.

**Analysis.** As shown in the Table 5, RSIR achieves consistent and significant improvements across all five metrics on all datasets.

- **Precision & F1-score:** The simultaneous increase in Precision and Recall (and consequently F1-score) is encouraging. In data augmentation scenarios, a common risk is introducing noise that might boost Recall (by covering more items) but degrade Precision (by recommending irrelevant items). The observed gains in Precision@10 (e.g., from 0.0080 to 0.0087 on Amazon-Toys) confirm that RSIR's fidelity control mechanism effectively filters out noise, ensuring that the densified signals remain highly relevant to user interests.

- **MRR:** The improvement in MRR (e.g., +11.1% on Yelp, from 0.0126 to 0.0140) indicates that RSIR not only retrieves relevant items but also ranks the first ground-truth item higher in the rank list. This suggests that the landscape smoothing effect of RSIR helps the model distinguish fine-grained preference differences, leading to more accurate ranking.

These comprehensive results further validate the generalizability and robustness of the RSIR framework, demonstrating that the performance gains are not an artifact of a specific metric but reflect a fundamental improvement in recommendation quality.

Table 6: Performance Comparison on Different Datasets (Metrics @ 10). The best results are highlighted in **bold**.

| Dataset | Precision@10 | | F1-score@10 | | MRR@10 | | NDCG@10 | | Recall@10 | |
|---------|------|-------|------|-------|------|-------|------|-------|------|-------|
| | Base | +RSIR | Base | +RSIR | Base | +RSIR | Base | +RSIR | Base | +RSIR |
| amazon-toys | 0.0080 | **0.0087** | 0.0145 | **0.0158** | 0.0380 | **0.0396** | 0.0477 | **0.0508** | 0.0795 | **0.0872** |
| amazon-beauty | 0.0055 | **0.0058** | 0.0100 | **0.0105** | 0.0212 | **0.0219** | 0.0290 | **0.0303** | 0.0548 | **0.0578** |
| amazon-sport | 0.0047 | **0.0051** | 0.0086 | **0.0093** | 0.0210 | **0.0227** | 0.0271 | **0.0293** | 0.0474 | **0.0512** |
| yelp | 0.0037 | **0.0040** | 0.0068 | **0.0072** | 0.0126 | **0.0140** | 0.0183 | **0.0200** | 0.0371 | **0.0399** |

## C.4 RECURSIVE SELF-IMPROVING PERFORMANCE

Figures 6a and 6b illustrate the performance of Recursive self-improving (RSI) on the Amazon-Sport and Yelp datasets, showing how the quality of the data evolves over iterations. The horizontal axis corresponds to the number of iterations, while the vertical axis indicates NDCG@20 and Recall@20.

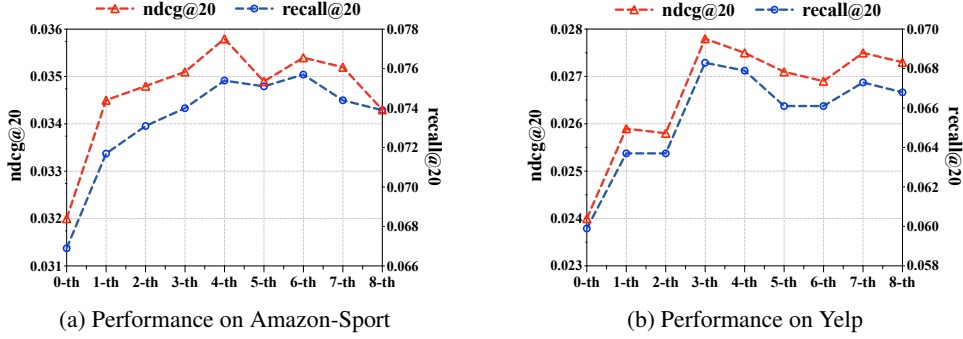

(a) Performance on Amazon-Sport    (b) Performance on Yelp

Figure 6: Performance of RSI Across Different Iterations on Amazon-Sport and Yelp.

Table 6 presents the performance of RSI on the Amazon-Sport and Yelp datasets across 8 iterations, along with the relative improvement compared to the previous round.

Table 7: Performance of RSI Across Multiple Iterations on Amazon-Sport and Yelp.(p-value $< 0.05$)

| | amazon-sport | | | | yelp | | | |
|---|---------|---------|---------|---------|---------|---------|---------|---------|
| | NDCG@10 | NDCG@20 | Recall@10 | Recall@20 | NDCG@10 | NDCG@20 | Recall@10 | Recall@20 |
| 0-th | 0.0271 | 0.0320 | 0.0474 | 0.0669 | 0.0183 | 0.0240 | 0.0371 | 0.0599 |
| 1-th | 0.0293 | 0.0345 | 0.0512 | 0.0717 | 0.0200 | 0.0259 | 0.0399 | 0.0637 |
| Improv | 8.12% | 7.81% | 8.02% | 7.17% | 9.29% | 7.92% | 7.55% | 6.34% |
| 2-th | 0.0294 | 0.0348 | 0.0517 | 0.0731 | 0.0198 | 0.0258 | 0.0399 | 0.0637 |
| Improv | 0.34% | 0.87% | 0.98% | 1.95% | -1.00% | -0.39% | 0.00% | 0.00% |
| 3-th | 0.0298 | 0.0351 | 0.0528 | 0.0740 | 0.0212 | 0.0278 | 0.0420 | 0.0683 |
| Improv | 1.36% | 0.86% | 2.13% | 1.23% | 7.07% | 7.75% | 5.26% | 7.22% |
| 4-th | 0.0301 | 0.0358 | 0.0527 | 0.0754 | 0.0210 | 0.0275 | 0.0419 | 0.0679 |
| Improv | 1.01% | 1.99% | -0.19% | 1.89% | -0.94% | -1.08% | -0.24% | -0.59% |
| 5-th | 0.0294 | 0.0349 | 0.0533 | 0.0751 | 0.0209 | 0.0271 | 0.0411 | 0.0661 |
| Improv | -2.33% | -2.51% | 1.14% | -0.40% | -0.48% | -1.45% | -1.91% | -2.65% |
| 6-th | 0.0300 | 0.0354 | 0.0540 | 0.0757 | 0.0206 | 0.0269 | 0.0408 | 0.0661 |
| Improv | 2.04% | 1.43% | 1.31% | 0.80% | -1.44% | -0.74% | -0.73% | 0.00% |
| 7-th | 0.0298 | 0.0352 | 0.0530 | 0.0744 | 0.0209 | 0.0275 | 0.0413 | 0.0673 |
| Improv | -0.67% | -0.56% | -1.85% | -1.72% | 1.46% | 2.23% | 1.23% | 1.82% |
| 8-th | 0.0286 | 0.0343 | 0.0513 | 0.0739 | 0.0211 | 0.0273 | 0.0423 | 0.0668 |
| Improv | -4.03% | -2.56% | -3.21% | -0.67% | 0.96% | -0.73% | 2.42% | -0.74% |

## C.5 HYPERPARAMETER ANALYSIS

Table 8 and 9 report the performance of our method on the *amazon-sport* dataset under different rank threshold $\tau$ and exploitation probability $p$, respectively. The evaluation metrics include NDCG@10, NDCG@20, Recall@10, and Recall@20.

Table 8: Performance of $\tau$ on *amazon-sport*.

| $\tau$ | NDCG@10 | NDCG@20 | Recall@10 | Recall@20 |
|------|---------|---------|-----------|-----------|
| *base* | *0.0271* | *0.0320* | *0.0474* | *0.0669* |
| 1 | 0.0277 | 0.0327 | 0.0469 | 0.0666 |
| 5 | 0.0288 | 0.0342 | 0.0507 | 0.0724 |
| 10 | 0.0280 | 0.0338 | 0.0495 | 0.0726 |
| 20 | 0.0293 | 0.0345 | 0.0512 | 0.0717 |
| 50 | 0.0286 | 0.0338 | 0.0505 | 0.0714 |

Table 9: Performance of $p$ on *amazon-sport*.

| $p$ | NDCG@10 | NDCG@20 | Recall@10 | Recall@20 |
|------|---------|---------|-----------|-----------|
| *base* | *0.0271* | *0.0320* | *0.0474* | *0.0669* |
| 0 | 0.0281 | 0.0336 | 0.0499 | 0.0716 |
| 0.3 | 0.0284 | 0.0331 | 0.0491 | 0.0679 |
| 0.5 | 0.0293 | 0.0345 | 0.0512 | 0.0717 |
| 0.7 | 0.0293 | 0.0347 | 0.0498 | 0.0714 |
| 0.9 | 0.0280 | 0.0327 | 0.0487 | 0.0677 |
| 1 | 0.0278 | 0.0330 | 0.0478 | 0.0683 |

### C.6 COMPATIBILITY WITH EXTERNAL KNOWLEDGE-ENHANCED MODELS

A prevalent approach to mitigating data sparsity is the incorporation of external knowledge, such as utilizing Large Language Models (LLMs) to generate item descriptions or employing Semantic IDs to capture hierarchical category information. In this section, we explore whether RSIR remains effective when the model already benefits from such external information.

We posit that RSIR is **orthogonal** to external knowledge integration. Methods leveraging external knowledge focus on enriching item representations with outside information, whereas RSIR focuses on maximizing the utility of the available interaction data through recursive self-generation. These distinct data augmentation perspectives allow the two strategies to work in parallel. To empirically validate this compatibility, we apply RSIR to a Semantic ID-based recommendation model (Wang et al., 2024a), which leverages external content hierarchies to map items into structured identifiers.

Table 10 presents the results on the *Amazon-Toys* dataset. The Semantic ID baseline (NDCG@10 = 0.0507) outperforms the standard ID-based SASRec (NDCG@10 = 0.0477, see Table 1), confirming that external knowledge effectively alleviates sparsity. Remarkably, applying RSIR on top of the Semantic ID model yields further significant improvements, boosting Recall@20 by **4.89%**.

This result demonstrates that RSIR is not redundant with external knowledge. Even when the model possesses rich, content-aware representations, RSIR's recursive mechanism can still further refine the model's performance by densifying the training data. Thus, RSIR can be seamlessly combined with knowledge-enhanced architectures.

Table 10: Performance comparison with semantic IDs on amazon-toys.

| Method | amazon-toys | | | |
|--------|---------|---------|-----------|-----------|
| | NDCG@10 | NDCG@20 | Recall@10 | Recall@20 |
| semantic id | 0.0507 | 0.0579 | 0.0837 | 0.1124 |
| + RSIR | **0.0518** | **0.0594** | **0.0877** | **0.1179** |
| Improv. | 2.17% | 2.59% | 4.78% | 4.89% |

## D DETAILED COMPUTATIONAL COMPLEXITY ANALYSIS

In this section, we provide a formal derivation of the time complexity for the RSIR framework.

### D.1 NOTATIONS AND PRELIMINARIES

We define the following notations for the complexity analysis:

- $N_k$: The number of sequences in the training dataset at iteration $k$.
- $L$: The maximum length of user sequences.
- $d$: The hidden state dimension of the model.
- $|\mathcal{V}|$: The size of the item vocabulary.
- $m$: The number of generation attempts per sequence.

- $K$: The total number of self-improvement iterations.

The backbone model $f_\theta$ typically consists of self-attention layers. The complexity for a forward pass on a single sequence, denoted as $\mathcal{C}_{\text{model}}$, is dominated by the attention mechanism:

$$\mathcal{C}_{\text{model}} \approx O(L^2 d + L d^2) \tag{6}$$

## D.2    COMPLEXITY DERIVATION

The RSIR process at iteration $k$ involves two distinct phases: (1) Training the model on $\mathcal{D}_k$, and (2) Generating the augmented set $\mathcal{D}'_{k+1}$.

**Phase 1: Model Training.**    At iteration $k$, the model is trained on $N_k$ sequences. Assuming convergence requires a constant number of epochs $E$, the training time complexity $\mathcal{T}_{\text{train}}^{(k)}$ is:

$$\mathcal{T}_{\text{train}}^{(k)} = O(E \cdot N_k \cdot \mathcal{C}_{\text{model}}) \tag{7}$$

Note that $N_k$ grows progressively. Let $\alpha$ be the effective expansion rate after fidelity filtering (where $0 \le \alpha \ll m$). Then $N_k \approx N_0(1+\alpha)^k$. Since $\alpha$ is strictly controlled by the fidelity threshold $\tau$, the dataset size remains within a manageable magnitude.

**Phase 2: Sequence Generation.**    For each of the $N_k$ sequences, we conduct $m$ generation trials. Let $L_e$ be the average *effective length* of the generated segments before the "Break" mechanism is triggered. For each generation step, the complexity includes:

1. **Inference:** Computing the hidden state, costing $\mathcal{C}_{\text{model}}$.

2. **Fidelity Check:** Calculating the dot product to rank candidates. A naive linear scan costs $O(d|\mathcal{V}|)$.

Thus, the generation complexity $\mathcal{T}_{\text{gen}}^{(k)}$ is:

$$\mathcal{T}_{\text{gen}}^{(k)} = O\left(N_k \cdot m \cdot L_e \cdot (\mathcal{C}_{\text{model}} + d|\mathcal{V}|)\right) \tag{8}$$

**Total Complexity.**    Summing over $K$ iterations, the total time complexity $\mathcal{T}_{\text{total}}$ is:

$$\mathcal{T}_{\text{total}} = \sum_{k=0}^{K-1} O\left(N_k \cdot (E \cdot \mathcal{C}_{\text{model}} + m \cdot L_e \cdot (\mathcal{C}_{\text{model}} + d|\mathcal{V}|))\right) \tag{9}$$

Given that the number of iterations $K$ is a small constant and the dataset expansion is strictly bounded, the cumulative time complexity scales linearly with respect to the initial dataset size $N_0$, vocabulary size $V$ and generation length $L_e$, ensuring the framework remains computationally scalable.

## D.3    OPTIMIZATION AND SCALABILITY

To address potential concerns regarding scalability on large datasets, we highlight two key properties:

**1. Effective Length Reduction ($L_e \ll L$):** The fidelity control mechanism serves as an early-stopping regularizer. If a generated item deviates from the user's preference manifold, the generation breaks immediately. This ensures that $L_e$ remains small, significantly reducing the multiplicative constant in $\mathcal{T}_{\text{gen}}$.

**2. Sub-linear Fidelity Check:** The term $d|\mathcal{V}|$ represents a Maximum Inner Product Search (MIPS) problem(Shrivastava & Li, 2014). By employing approximate retrieval structures (e.g., HNSW(Malkov & Yashunin, 2018) or Tree-based indexing(Ram & Gray, 2012)), the complexity of the fidelity check reduces from linear $O(d|\mathcal{V}|)$ to logarithmic $O(d \log |\mathcal{V}|)$. This ensures that the cost does not explode even when the vocabulary size $|\mathcal{V}|$ is extremely large.

## D.4 Empirical Runtime Analysis

To validate our theoretical complexity analysis, we conduct an empirical runtime comparison against competitive generative baselines, including DR4SR(Yin et al., 2024) and ASReP(Liu et al., 2021b). The experiments are conducted on the same hardware environment to ensure fairness. The results are reported in Table 12.

**Generation Efficiency.** As shown in Table 12, RSIR demonstrates a substantial advantage in the data generation phase. Specifically, RSIR is approximately **18× faster** than the pattern-based method DR4SR (3m vs. 68m) and **5× faster** than ASReP. This empirical result strongly corroborates our theoretical assertion: by utilizing the backbone recommendation model itself as the generator and employing the "Break" mechanism to constrain the effective generation length ($L_e$), RSIR avoids the heavy computational burden associated with complex external generators.

**Training Efficiency.** A striking observation from Table 12 is that the retraining time of RSIR (2m16s) is comparable to, or even slightly faster than, the training time of the Base model (2m34s), despite the increased data volume. This counter-intuitive result empirically supports our theoretical insight regarding **implicit regularization** (Section 4). The high-fidelity synthetic data generated by RSIR smooths the optimization landscape, enabling the optimizer to converge more quickly to a robust solution. Compared to baselines like DR4SR (approx. 10m training time), RSIR maintains orders-of-magnitude superior efficiency, confirming that full retraining is computationally feasible and efficient in our framework.

**Deployment Potential and Acceleration.** It is important to note that the reported generation time for RSIR was measured using a sequential implementation **without parallelization strategies**. Consequently, significant room for acceleration remains via standard engineering optimizations. We provide a preliminary exploration and validation of such parallel strategies in **Appendix D.5**. Furthermore, the data generation phase is decoupled from training and can be executed **offline**. Combined with our findings in **Section 5.4**—where weak models can effectively instruct stronger ones—practitioners can utilize a lightweight, high-throughput model for offline data generation to efficiently train a large-scale production model, maximizing industrial viability.

## D.5 Scalability Optimization via Clustering-based Retrieval

A primary challenge in deploying RSIR to large-scale industrial systems is the computational cost of the fidelity check (Appendix D.2), which theoretically requires scanning the entire item vocabulary $|\mathcal{V}|$. To validate the feasibility of accelerating this process without compromising performance, we propose a Clustering-based Approximate Retrieval strategy.

**Implementation Strategy.** We adopt a two-stage retrieval approach to prune the candidate space:

1. **Clustering:** We partition the global item set into $C$ clusters (Liu et al., 2024) and compute a centroid for each cluster.
2. **Approximate Search:** During the generation phase, instead of scanning all items, the model first calculates the similarity between the current context and cluster centroids to select the top-$k$ most relevant clusters. The candidate pool $\mathcal{V}_{sub}$ is then restricted to items within these clusters.

This strategy reduces the complexity of the fidelity check from linear $O(|\mathcal{V}|)$ to sub-linear, making it scalable to millions of items.

**Empirical Validation.** We simulated this strategy on the Amazon-Sport and Yelp datasets. The results are presented in Table 11.

RSIR-Cluster consistently outperforms the Base (SASRec) model by a significant margin. It also achieves performance highly comparable to the exact RSIR implementation. The performance gap is negligible (e.g., $< 1.7\%$ drop in Recall@10 on Amazon-Sport), and in some cases (e.g., NDCG@10 on Yelp), RSIR-Cluster even marginally outperforms the exact version. This suggests that clustering may act as an additional denoising filter by excluding irrelevant items.

Table 11: Ablation study of clustering module on amazon-sport and yelp datasets.

| Method | amazon-sport | | | | yelp | | | |
|---|---|---|---|---|---|---|---|---|
| | NDCG@10 | NDCG@20 | Recall@10 | Recall@20 | NDCG@10 | NDCG@20 | Recall@10 | Recall@20 |
| SASRec | 0.0271 | 0.0320 | 0.0474 | 0.0669 | 0.0183 | 0.0240 | 0.0371 | 0.0599 |
| + RSIR | **0.0293** | **0.0345** | **0.0512** | 0.0717 | 0.0200 | **0.0259** | **0.0399** | **0.0637** |
| + RSIR-Cluster | 0.0283 | 0.0340 | 0.0503 | **0.0729** | **0.0201** | 0.0258 | 0.0397 | 0.0635 |

Table 12: Time Efficiency Comparison of Different Methods (measured on Amazon-Toys). Note that RSIR involves **retraining from scratch**, yet remains highly efficient.

| Phase | Base | RSIR | DR4SR | ASReP |
|---|---|---|---|---|
| Data Generation Phase | - | **3m45.922s** | 68m48.733s | 20m13.968s |
| Training Phase | 2m34.605s | **2m16.159s** | 10m40.349s | 3m44.264s |

These results confirm that approximate retrieval effectively captures the on-manifold candidates required for self-improvement while drastically reducing the search space, thereby resolving the deployment bottleneck associated with large vocabularies.

## E  THEORETICAL ANALYSIS AND PROOFS

In this section, we provide the formal proofs supporting the theoretical claims made in Section 4. We first derive the geometric form of the implicit regularizer introduced by RSIR (Section E.1) and then provide the derivation for the recursive error bound and convergence conditions (Section E.2).

### E.1  PROOF OF MANIFOLD TANGENTIAL GRADIENT PENALTY

**Problem Statement:** We aim to characterize the implicit regularization term $\Omega(\theta; \theta_k)$ imposed by minimizing the loss on the generated dataset $D'_{k+1}$.

**Assumption 1 (Manifold Hypothesis):** User preferences lie on a low-dimensional manifold $\mathcal{M}$ embedded in the high-dimensional item space(Belkin et al., 2006).

**Assumption 2 (Local Consistency):** A generated sequence $s' \in D'_{k+1}$ is a local neighbor of a real sequence $s_{ctx}$, such that the difference vector $v = s' - s_{ctx}$ lies approximately in the tangent space $T_s \mathcal{M}$ of the manifold.

**Derivation:** The regularization effect arises from enforcing consistency between the model's predictions on the context $s_{ctx}$ and its generated neighbor $s'$. We define the regularization objective as the expected squared difference:

$$\Omega(\theta) = \mathbb{E}_{s_{ctx} \sim \mathcal{D}, s' \sim P(\cdot|s_{ctx})} \left[ \|f_\theta(s') - f_\theta(s_{ctx})\|^2 \right] \tag{10}$$

Using a first-order Taylor expansion of $f_\theta(s')$ around $s_{ctx}$:

$$f_\theta(s') \approx f_\theta(s_{ctx}) + \nabla_s f_\theta(s_{ctx})^\top (s' - s_{ctx}) \tag{11}$$

Let $v = s' - s_{ctx}$. Substituting this into the objective:

$$\Omega(\theta) \approx \mathbb{E}\left[ \|\nabla_s f_\theta(s_{ctx})^\top v\|^2 \right] = \mathbb{E}\left[ v^\top \nabla_s f_\theta \nabla_s f_\theta^\top v \right] \tag{12}$$

Using the trace trick ($x^\top A x = \text{Tr}(A x x^\top)$):

$$\Omega(\theta) \approx \text{Tr}\left( \nabla_s f_\theta \nabla_s f_\theta^\top \mathbb{E}[v v^\top] \right) \tag{13}$$

Since RSIR explores the local neighborhood of the user's preference manifold, the covariance of the perturbation $v$ is proportional to the projection matrix $\mathcal{P}_\mathcal{M}$ onto the tangent space $T_s \mathcal{M}$. Letting $\mathbb{E}[v v^\top] = \sigma^2 \mathcal{P}_\mathcal{M}$:

$$\Omega(\theta) \propto \text{Tr}\left( \nabla_s f_\theta \nabla_s f_\theta^\top \mathcal{P}_\mathcal{M} \right) = \nabla_s f_\theta^\top \mathcal{P}_\mathcal{M} \nabla_s f_\theta \tag{14}$$

Since $\mathcal{P}_{\mathcal{M}}$ is an orthogonal projection matrix (idempotent, $\mathcal{P}_{\mathcal{M}}^{\top}\mathcal{P}_{\mathcal{M}} = \mathcal{P}_{\mathcal{M}}$), we have:

$$\nabla_s f_\theta^{\top} \mathcal{P}_{\mathcal{M}}^{\top} \mathcal{P}_{\mathcal{M}} \nabla_s f_\theta = \|\mathcal{P}_{\mathcal{M}} \nabla_s f_\theta\|^2 \equiv \|\nabla_{\mathcal{M}} f_\theta\|^2 \tag{15}$$

**Conclusion:** The implicit regularizer minimizes $\|\nabla_{\mathcal{M}} f_\theta\|^2$, the norm of the gradient projected onto the manifold. This confirms that RSIR enforces smoothness specifically along valid user preference trajectories. $\square$

### E.2 Recursive Error Bound and Convergence Analysis

We define $\mathcal{E}(\theta_k)$ as the generalization error of the model at iteration $k$. The dataset at iteration $k+1$ is a mixture of the original sparse data (ratio $1 - \lambda$) and the generated dense data (ratio $\lambda$).

**Theorem 1 (Recursive Error Bound).** *Under the RSIR framework, the error dynamics follow the inequality:*

$$\mathcal{E}(\theta_{k+1}) \leq (1 - \lambda)\mathcal{E}_0 + \lambda\left[(1 - \tilde{p}_k)\rho\mathcal{E}(\theta_k) + \tilde{p}_k\mathcal{E}_{\max}\right] \tag{16}$$

*where $\rho < 1$ is the contraction rate from valid data expansion, $\tilde{p}_k$ is the effective noise rate (fidelity leakage), and $\mathcal{E}_{max}$ is the maximum bounded loss.*

*Proof.* The total error is the convex combination of errors on the original and generated distributions.

1. On the original data $D_0$, the error is bounded by the baseline error $\mathcal{E}_0$.

2. The generated data $D'_{k+1}$ consists of:

   - **Valid Sequences (True Positives):** Proportion $(1 - \tilde{p}_k)$. These sequences reside on the true manifold. By the expansion-contraction principle of self-training, optimizing on these samples contracts the error relative to the previous iteration: $\mathcal{E}_{\text{valid}} \leq \rho\mathcal{E}(\theta_k)$ Wei et al. (2020).
   - **Invalid Sequences (False Positives):** Proportion $\tilde{p}_k$. These are off-manifold noise. The error is bounded by the loss function's maximum value: $\mathcal{E}_{\text{invalid}} \leq \mathcal{E}_{\max}$.

   Combining these terms yields the theorem statement.

$\square$

**Corollary (Stability Condition).** For the system to self-improve ($\mathcal{E}(\theta_{k+1}) < \mathcal{E}(\theta_k)$), the leakage rate $\tilde{p}_k$ must satisfy:

$$\tilde{p}_k < \frac{\mathcal{E}(\theta_k)(1 - \lambda\rho) - (1 - \lambda)\mathcal{E}_0}{\lambda(\mathcal{E}_{\max} - \rho\mathcal{E}(\theta_k))} \tag{17}$$

This upper bound is the **Breakdown Point**. If the fidelity control is too loose ($\tau$ is too high), $\tilde{p}_k$ exceeds this threshold, causing error divergence(Kumar et al., 2020). Conversely, a strict $\tau$ ensures $\tilde{p}_k \approx 0$, leading to monotonic convergence.

Assuming the noise $\tilde{p}_k$ is negligible due to a strict $\tau$, the dynamics simplify to a linear contraction mapping. The error converges to a fixed limit $\mathcal{E}^*$:

$$\lim_{k \to \infty} \mathcal{E}(\theta_k) = \frac{(1 - \lambda)\mathcal{E}_0}{1 - \lambda\rho}. \tag{18}$$

Since $\rho < 1$, it follows that $\mathcal{E}^* < \mathcal{E}_0$, proving that RSIR achieves a lower error than standard supervised learning. However, as $\mathcal{E}(\theta_k)$ approaches $\mathcal{E}^*$, the term $\rho\mathcal{E}(\theta_k)$ shrinks, meaning the marginal gain from each iteration diminishes.

Nevertheless, $\tilde{p}_k$ is never exactly zero, so $\tilde{p}_k > 0$. The term $\lambda\tilde{p}_k\mathcal{E}_{\max}$ acts as an irreducible noise floor. In early iterations, the improvement from contraction ($\rho\mathcal{E}(\theta_k)$) dominates the noise. However, as the model improves ($\mathcal{E}(\theta_k)$ becomes small), the relative impact of the leakage noise $\tilde{p}_k\mathcal{E}_{\max}$ increases. If the noise term eventually outweighs the shrinking contraction benefit, the performance curve may show a slight degradation after optimal iterations. This theoretical insight underscores

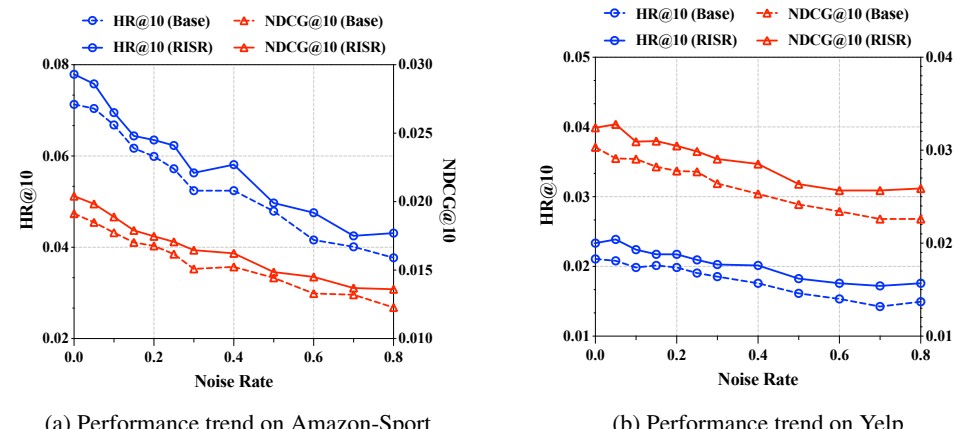

(a) Performance trend on Amazon-Sport  (b) Performance trend on Yelp

Figure 7: Performance comparison under different noise ratios. While performance naturally degrades with increased noise, RSIR consistently maintains a significant lead over the Base model.

the importance of our fidelity-based quality control: it is the crucial mechanism for suppressing $\tilde{p}_k$ and maintaining the noise floor below the contraction benefit. □

## F   ROBUSTNESS ANALYSIS UNDER DATA NOISE

In real-world scenarios, user interaction logs often contain noise—accidental clicks or irrelevant interactions—that can mislead the recommender system(Wang et al., 2021). A critical concern is whether the self-improving loop of RSIR might amplify such noise, leading to error propagation(Arazo et al., 2020).

To evaluate the robustness of RSIR, we conducted a controlled experiment by injecting varying ratios of random noise into the training data. Specifically, for each user sequence, we randomly inserted items from the global item set with a noise ratio $\eta \in [0, 0.8]$. We compared the performance of RSIR against the Base model across two datasets, Amazon-Sport and Yelp.

**Results and Analysis.**   Figure 7 illustrates the performance trends, and Table 13 details the numerical results. We observe two key findings:

1. **Consistent Superiority:** As expected, the absolute performance of both the Base model and RSIR declines as the noise ratio increases. However, as shown in Figure 7, RSIR consistently stays above the Base baseline across the entire noise spectrum (from 0% to 80%), demonstrating that our framework does not collapse even under severe data contamination.

2. **Increased Relative Gain in Noisy Environments:** Crucially, Table 13 reveals that the *relative improvement* brought by RSIR tends to increase as the data becomes noisier.
   - On **Amazon-Sport**, at a low noise ratio ($\eta = 0$), the improvement in Recall@10 is **8.02%**. When noise increases to extreme levels ($\eta = 0.8$), the improvement jumps to **14.93%**.
   - Similarly, on **Yelp**, the improvement in Recall@10 rises from **7.55%** (at $\eta = 0$) to **16.42%** (at $\eta = 0.8$).

**Discussion.**   These results strongly support our theoretical insight regarding **Implicit Regularization** (Section 4). Random noise typically constitutes off-manifold perturbations(Verma et al., 2019). The fidelity-based quality control mechanism in RSIR effectively filters out these random, low-probability interactions during the generation phase, preventing them from being reinforced in the self-training loop. By selectively densifying the valid, on-manifold user trajectories, RSIR acts as a **denoising filter**, enabling the model to learn robust preferences even when the original signal is heavily corrupted.

Table 13: Performance robustness comparison under varying noise rates. The best performance in each comparison is highlighted in **bold**. Improvement percentages are shaded in gray.

| Noise | Method | amazon-sport | | | | yelp | | | |
|---|---|---|---|---|---|---|---|---|---|
| | | NDCG@10 | NDCG@20 | Recall@10 | Recall@20 | NDCG@10 | NDCG@20 | Recall@10 | Recall@20 |
| 0 | Base | 0.0271 | 0.0320 | 0.0474 | 0.0669 | 0.0183 | 0.0240 | 0.0371 | 0.0599 |
| | +RSIR | **0.0293** | **0.0345** | **0.0512** | **0.0717** | **0.0200** | **0.0259** | **0.0399** | **0.0637** |
| | Improv. | 8.12% | 7.81% | 8.02% | 7.17% | 9.29% | 7.92% | 7.55% | 6.34% |
| 0.05 | Base | 0.0268 | 0.0319 | 0.0455 | 0.0656 | 0.0181 | 0.0242 | 0.0355 | 0.0598 |
| | +RSIR | **0.0286** | **0.0340** | **0.0495** | **0.0709** | **0.0204** | **0.0264** | **0.0404** | **0.0643** |
| | Improv. | 6.72% | 6.58% | 8.79% | 8.08% | 12.71% | 9.09% | 13.80% | 7.53% |
| 0.1 | Base | 0.0256 | 0.0302 | 0.0432 | 0.0615 | 0.0174 | 0.0232 | 0.0354 | 0.0583 |
| | +RSIR | **0.0265** | **0.0313** | **0.0467** | **0.0658** | **0.0193** | **0.0250** | **0.0379** | **0.0604** |
| | Improv. | 3.52% | 3.64% | 8.10% | 6.99% | 10.92% | 7.76% | 7.06% | 3.60% |
| 0.15 | Base | 0.0239 | 0.0282 | 0.0411 | 0.0581 | 0.0176 | 0.0234 | 0.0343 | 0.0574 |
| | +RSIR | **0.0248** | **0.0299** | **0.0437** | **0.0640** | **0.0188** | **0.0250** | **0.0380** | **0.0627** |
| | Improv. | 3.77% | 6.03% | 6.33% | 10.15% | 6.82% | 6.84% | 10.79% | 9.23% |
| 0.2 | Base | 0.0233 | 0.0275 | 0.0403 | 0.0569 | 0.0174 | 0.0223 | 0.0337 | 0.0534 |
| | +RSIR | **0.0245** | **0.0291** | **0.0424** | **0.0608** | **0.0188** | **0.0244** | **0.0373** | **0.0594** |
| | Improv. | 5.15% | 5.82% | 5.21% | 6.85% | 8.05% | 9.42% | 10.68% | 11.24% |
| 0.25 | Base | 0.0224 | 0.0263 | 0.0385 | 0.0543 | 0.0168 | 0.0222 | 0.0336 | 0.0548 |
| | +RSIR | **0.0241** | **0.0289** | **0.0412** | **0.0601** | **0.0182** | **0.0236** | **0.0365** | **0.0582** |
| | Improv. | 7.59% | 9.89% | 7.01% | 10.68% | 8.33% | 6.31% | 8.63% | 6.20% |
| 0.3 | Base | 0.0208 | 0.0248 | 0.0353 | 0.0512 | 0.0164 | 0.0218 | 0.0319 | 0.0535 |
| | +RSIR | **0.0221** | **0.0265** | **0.0394** | **0.0567** | **0.0177** | **0.0229** | **0.0354** | **0.0564** |
| | Improv. | 6.25% | 6.85% | 11.61% | 10.74% | 7.93% | 5.05% | 10.97% | 5.42% |
| 0.4 | Base | 0.0208 | 0.0246 | 0.0357 | 0.0508 | 0.0157 | 0.0208 | 0.0304 | 0.0511 |
| | +RSIR | **0.0227** | **0.0267** | **0.0387** | **0.0548** | **0.0176** | **0.0228** | **0.0347** | **0.0556** |
| | Improv. | 9.13% | 8.54% | 8.40% | 7.87% | 12.10% | 9.62% | 14.14% | 8.81% |
| 0.5 | Base | 0.0193 | 0.0226 | 0.0333 | 0.0465 | 0.0146 | 0.0191 | 0.0289 | 0.0471 |
| | +RSIR | **0.0199** | **0.0239** | **0.0346** | **0.0504** | **0.0162** | **0.0212** | **0.0318** | **0.0517** |
| | Improv. | 3.11% | 5.75% | 3.90% | 8.39% | 10.96% | 10.99% | 10.03% | 9.77% |
| 0.6 | Base | 0.0172 | 0.0204 | 0.0299 | 0.0429 | 0.0140 | 0.0190 | 0.0279 | 0.0476 |
| | +RSIR | **0.0192** | **0.0228** | **0.0335** | **0.0477** | **0.0157** | **0.0204** | **0.0309** | **0.0499** |
| | Improv. | 11.63% | 11.76% | 12.04% | 11.19% | 12.14% | 7.37% | 10.75% | 4.83% |
| 0.7 | Base | 0.0167 | 0.0200 | 0.0296 | 0.0426 | 0.0132 | 0.0174 | 0.0268 | 0.0437 |
| | +RSIR | **0.0175** | **0.0211** | **0.0311** | **0.0452** | **0.0154** | **0.0201** | **0.0309** | **0.0495** |
| | Improv. | 4.79% | 5.50% | 5.07% | 6.10% | 16.67% | 15.52% | 15.30% | 13.27% |
| 0.8 | Base | 0.0159 | 0.0189 | 0.0268 | 0.0386 | 0.0137 | 0.0182 | 0.0268 | 0.0446 |
| | +RSIR | **0.0177** | **0.0209** | **0.0308** | **0.0437** | **0.0157** | **0.0205** | **0.0312** | **0.0506** |
| | Improv. | 11.32% | 10.58% | 14.93% | 13.21% | 14.60% | 12.64% | 16.42% | 13.45% |

## G    QUANTITATIVE EVALUATION OF GENERATED DATA

In addition to evaluating recommendation performance using metrics such as Hit Rate (HR) and Normalized Discounted Cumulative Gain (NDCG), we further assess the intrinsic properties of the generated data using Approximate Entropy (ApEn) (Pincus, 1991), a statistical measure that quantifies the regularity and unpredictability of sequences. In the context of recommender systems, ApEn can capture the complexity and diversity of individual users' interaction sequences, providing complementary insights beyond conventional accuracy-based metrics.

In our implementation, the ApEn is computed as follows: Given a user interaction sequence $s_u$ of length $N$, the embedding dimension is $m$, and a similarity tolerance $r$. We first construct an $m$-dimensional subsequence vector: $v_k^m = [i_k, i_{k+1}, ..., i_{k+m-1}]$ for $k = 1, ..., N - m + 1$. The distance between two subsequences is measured using the Chebyshev distance:

$$d[v_k^m, v_j^m] = \max_{0 \le q < m} |x_{k+q} - x_{j+q}|$$

The similarity between subsequences under the tolerance $r$ is then calculated as:

$$C_k^m(r) = \frac{|\{j | d[v_k^m, v_j^m] \le r\}|}{N - m + 1}$$

Next, the average logarithmic similarity of all length-$m$ subsequences is computed:

$$\Phi^m(r) = \frac{1}{N-m+1} \sum_{k=1}^{N-m+1} \ln C_k^m(r)$$

Finally, the Approximate Entropy of the user sequence is defined as:

$$ApEn(m, r; s_u) = \Phi^m(r) - \Phi^{m+1}(r)$$

In our implementation, we set $r = 0$ due to the unique nature of recommended items, where similar item IDs may represent entirely different products. To align the measure with the conventional notion of diversity, we use the reciprocal: $ApEn' = 1/ApEn$, following Shen et al. (2024). For each user's interaction sequence, a higher ApEn value reflects greater complexity and information density in a sequence, making it a richer source of information for training the model.

# H    SEQUENTIAL RECOMMENDATION PARADIGM

Sequential recommendation aims to model the evolving preferences of users by predicting their next interactions based on historical behaviors.

Given an input sequence $s_u = (i_1, i_2, ..., i_{|s_u|})$ at step $t$, sequential recommendation models learn the conditional probability distribution $p(i_t|i_{<t})$, where $i_{<t} = (i_1, i_2, ..., i_{t-1})$ represents the subsequence before the $t$-th item.

To model the conditional distribution $p(i_t|i_{<t})$, the prefix sequence $i_{<t}$ is first mapped into a sequence of embeddings $\mathbf{E}_{<t} = (e_1, e_2, ..., e_{t-1})$ through an embedding layer. Then the sequential encoder(Transformer, RNN, CNN, or other architectures) $f_\theta(\cdot)$ generates a context representation $\mathbf{h_t}$ for position $t$:

$$\mathbf{h}_t = f_\theta(\mathbf{E}_{<\mathbf{t}})$$

The probability of each candidate item $v \in \mathcal{V}$ will be computed via an inner product operation or other scoring function between $\mathbf{h}_t$ and the item embedding $\mathbf{e}_v$, and the final probability will be normalized with a softmax:

$$p(i_t = v|i_{<t}) = \frac{\exp\left(\mathbf{h}_t^T \mathbf{e}_v\right)}{\sum_{v \in \mathcal{V}} \exp\left(\mathbf{h}_t^T \mathbf{e}_v\right)}$$

At inference time, the recommender outputs the item with the highest predicted probability:

$$i_t = \arg\max_{v \in \mathcal{V}} p(i_t = v|i_{<t})$$

For training, the model is optimized using a sampled softmax cross-entropy loss. Given the true target item $v^+$ at position $t$ and a sampled subset of candidate items $\mathcal{C} \subseteq \mathcal{V}$, the loss at step $t$ is calculated as:

$$\mathcal{L}_t(\theta) = -\log \frac{\exp\left(\mathbf{h}_t^T \mathbf{e}_{v^+}\right)}{\sum_{v \in \mathcal{C}} \exp\left(\mathbf{h}_t^T \mathbf{e}_v\right)}$$

The overall training objective sums (or averages) the per-position losses across the entire sequence:

$$\mathcal{L}(\theta) = \frac{1}{|s_u|} \sum_{t=1}^{|s_u|} \mathcal{L}_t(\theta).$$

Other loss functions, such as full softmax cross-entropy, Bayesian Personalized Ranking (BPR), or pairwise hinge loss, can also be used for sequential recommendation, but in our experiments, we adopt the sampled softmax loss to match our method design.

A widely used instantiation of this general framework is SASRec(Kang & McAuley, 2018), which adopts a stack of self-attention layers as $f_\theta(\cdot)$ to model long-range dependencies within $i_{<t}$. Other models may replace the self-attention block with GRUs, CNNs, or graph neural networks, but the above conditional modeling and factorization remain the same.

## I    THE USE OF LARGE LANGUAGE MODELS

All technical aspects of this work, including the conception of the method, the design of experiments, and the implementation of algorithms, were conceived and executed independently by the authors without the involvement of large language models. During the writing process, all sections of the manuscript were written by the authors themselves, and large language models were used only to improve the wording of text that had already been completed.

