# OpenReview forum: "Can Recommender Systems Teach Themselves? A Recursive Self-Improving Framework with Fidelity Control"
_ICLR.cc/2026/Conference — Submitted to ICLR 2026_

### Official Review · Reviewer_jDuY · 2025-10-27

**Soundness:** 2
**Presentation:** 3
**Contribution:** 2
**Rating:** 4
**Confidence:** 4

**Summary:**

This paper proposes the RSIR framework to address data sparsity in recommender systems, enabling models to self-improve via a closed loop—generating interaction sequences, filtering them with fidelity control, and retraining on enriched data without external reliance. Theoretically, RSIR acts as a data-driven implicit regularizer to smooth the optimization landscape .

**Strengths:**

1. RSIR significantly addresses the core issue of extreme data sparsity in recommender systems.

2. Unlike traditional methods that rely on external curated data (expensive and domain-specific) or teacher models (risk of distribution mismatch), RSIR operates in a closed loop. It leverages the current model’s own understanding of user preferences to generate and filter synthetic sequences, eliminating reliance on external resources .

**Weaknesses:**

1. The iterative self-improvement loop involves repeated steps of sequence generation (generating m synthetic sequences per user), fidelity filtering, and model retraining. For large datasets or multiple iterations, this process increases computational costs — e.g., generating 20 synthetic sequences per user (m=20) and retraining models from scratch each iteration adds significant time and resource burdens .

2. Why does each iteration require the recommender system to be retrained from scratch? Why not employ incremental training? In practical industrial settings, incremental training is the more common practice, since retraining from scratch is prohibitively expensive.

**Questions:**

1. In each iteration, when randomly sampling from the candidate item set, does the set include all items from the training split or all items from the entire dataset?

2. In real-world deployments, the candidate pool is enormous. With random sampling, is there a risk of not retrieving items that satisfy the quality-control criteria? Alternatively, would many sampling attempts be required to find a suitable item, thereby greatly increasing computational cost?

---

> ### Author Response · Authors · 2025-11-22
>
> We sincerely thank you for recognizing that RSIR **"significantly addresses the core issue of extreme data sparsity"** and operates in a **valuable** "closed loop" without external dependencies. We understand your concerns regarding industrial scalability and retraining costs. We have added specific experiments (Fine-Tuning in **Table 1** and Approximate Retrieval in **Appendix D.5 on page 20**) to address these points directly.
>
> > **W1 & W2: Computational costs and Retraining vs. Incremental Training**
>
> We fully agree that full retraining can be costly in industrial settings. We have addressed this with two key findings:
>
> 1.  **Compatibility with Incremental Learning (Table 1):** To address your concern about "retraining from scratch," we implemented **RSIR-FT (Fine-Tuning)**, where we only fine-tune the model from the previous iteration.
>     *   **Result:** As reported in **Table 1**, `RSIR-FT` achieves performance comparable to (and sometimes better than) full retraining (e.g., **0.0594** Recall@10 on Amazon-Beauty vs. 0.0578).
>     *   **Conclusion:** This indicates that RSIR can be integrated into standard incremental training pipelines commonly used in industry.
>
> 2.  **Accelerated Convergence (Table 12 on page 21):** Counter-intuitively, even when retraining from scratch, RSIR often **accelerates convergence**. As shown in **Table 12 (Appendix D.4)**, retraining RSIR takes **2m 16s**, which is actually faster than training the Base model on sparse data (2m 34s). This supports our theory (**Section 4.1**) that RSIR smooths the optimization landscape, allowing the optimizer to find robust solutions with fewer epochs.
>
> > **Q1: Scope of the candidate item set during sampling**
>
> *   The candidate pool consists strictly of the **global item set $\mathcal{V}$ present in the training split**.
> *   **No Leakage:** Items that appear *only* in the validation or test sets are strictly excluded to prevent data leakage.
>
> > **Q2: Sampling efficiency in large item spaces**
>
> This is an excellent engineering question. We address the risk of "missed items" and "high computational cost" in two ways:
>
> 1.  **Model-Guided Sampling (Not Blind Randomness):** We do not perform uniform random sampling. Instead, we sample from the **top-$k$ of the model's predicted probability distribution**.
>     *   **Why this works:** Since the model (even a weak one) has learned basic collaborative patterns, its top predictions are highly likely to be relevant. Therefore, the sampled items frequently pass the fidelity check without requiring many rejected attempts. This keeps the generation cost low.
>
> 2.  **Scalability via Clustering (Appendix D.5 on page 20):** For massive industrial datasets where the vocabulary $|\mathcal{V}|$ is in the millions, the bottleneck is the fidelity check (ranking). We introduced a **Clustering-based Approximate Retrieval** strategy.
>     *   **Method:** We partition items into clusters and first retrieve the top-$k$ relevant *clusters* before sampling items.
>     *   **Result:** **Table 11** on page 21 shows that this strategy reduces the search complexity from linear $O(|\mathcal{V}|)$ to **sub-linear**, with negligible performance loss. This ensures RSIR scales efficiently to enormous item pools.

---

### Official Review · Reviewer_Rqbe · 2025-10-30

**Soundness:** 2
**Presentation:** 3
**Contribution:** 2
**Rating:** 4
**Confidence:** 3

**Summary:**

This paper addresses the challenge of extreme user interaction sparsity in recommender systems, which leads to optimization difficulty and poor generalization. The authors propose a Recursive Self-Improving Recommendation (RSIR) framework that enables a model to continuously improve itself without external data or teacher models. RSIR operates by generating plausible user–item interaction sequences with the current model, filtering them through a fidelity-based quality control mechanism to retain only high-quality samples aligned with real user preferences, and using these to train the next-generation model. Experimental results on multiple benchmarks and architectures demonstrate consistent and cumulative performance improvements across iterations.

**Strengths:**

* The user sparsity issue in recommender systems is important and highly relevant to the community.
* The proposed three-stage “generate–filter–retrain” pipeline is conceptually simple, broadly applicable, and uses transparent hyperparameters, which facilitates reproducibility.
* The authors conduct extensive experiments, showing improvements on multiple datasets and model backbones, with multi-round benefits until convergence or saturation.

**Weaknesses:**

* The fidelity-based quality control mechanism is tightly coupled to the *future ground truth*, which may limit generalization and realism.

  * Specifically, the filter accepts a generated step only if, after insertion, at least one *real* future item remains highly ranked (≤ τ).
  * This effectively constrains the generation process to small perturbations around the known future sequence, making many “new sequences” closer to alternative prefixes of the same label rather than genuinely novel user trajectories.
  * It would strengthen the work if the authors could propose ways to address this issue or introduce alternative criteria that decouple fidelity assessment from future ground truth.
* The baseline coverage is relatively narrow. Current comparisons focus only on simple insertion and reordering augmentations, but stronger and more recent baselines—such as data regeneration, diffusion-based, or generative augmentation methods that actively synthesize sequential data—are missing.
* The theoretical discussion is mostly intuitive and lacks formal guarantees or conditions.

  * The idea of “implicit regularization” or “landscape smoothing” is interesting but remains qualitative; no formal bounds or conditions are provided to clarify when RSIR helps versus hurts.
  * Moreover, the technical components (especially *self-training*) have already been explored in other recommendation contexts. Since the novelty here largely lies in the generation criterion, a deeper and more rigorous theoretical analysis would be crucial to justify the claimed insights. I strongly suggest the authors expand or formalize the theoretical section in their rebuttal.
* Efficiency reporting is missing.
  The approach involves multi-round training and parameter searches, but no details on training time, computational cost, or resource consumption are provided. Such analysis would be helpful for assessing scalability and practical feasibility.

**Questions:**

Please refer to the weaknesses above, particularly regarding:

* How to mitigate the reliance of fidelity control on future ground truth;
* Whether stronger generative baselines can be added;
* How the theoretical section could be strengthened with formal analysis or guarantees;
* And what the training cost and computational overhead are for multi-round RSIR.

---

> ### Author Response · Authors · 2025-11-22
>
> We sincerely thank you for your constructive review. We appreciate that you found our pipeline "conceptually simple, broadly applicable" and our experiments "extensive." Your critique regarding **generative baselines**, **theoretical rigor**, and the **reliance on future ground truth** pinpointed the exact areas where our initial manuscript was lacking.
>
> In response, we have performed a major revision: we added **three state-of-the-art generative baselines** (Table 1), derived a **formal Recursive Error Bound** (**Appendix E** on page 21), and conducted a detailed **efficiency analysis** (**Appendix D** on page 18).
>
> > **W1 & Q1: Reliance of fidelity control on future ground truth**
>
> We appreciate this important observation. The current fidelity check ensures that generated sequences remain aligned with the user’s established interest space, which helps prevent error accumulation. We have clarified this design rationale in **Section 3.2** and **Appendix E.2 on page 22**.
>
> As shown in Section 5.5 and Figure 5b, RSIR-generated datasets consistently exhibit **higher entropy** than the original sparse logs (unlike simple augmentation methods like Insertion, which decrease entropy). This proves that despite the ground-truth constraint, RSIR introduces **complex, information-rich patterns** that are not present in the original data.
>
> To reduce dependence on future interactions, we will explore  some alternative criteria, including (1) using **model confidence** rather than future labels and (2) incorporating **auxiliary reward signals**. We view this as a promising direction and have noted it as a limitation and avenue for future work.
>
> > **W2 & Q2: Baseline coverage is too narrow**
>
> We agree that broader baselines strengthen the contribution. We have expanded our comparison to include **three advanced learnable data generation methods**: a sequence extension method **ASReP**, a Diffusion-based generation method **DiffuASR**, and a regeneration-based method **DR4SR**.
> *   **Performance:** As shown in the updated **Table 1**, RSIR consistently outperforms these complex baselines (e.g., beating DR4SR by **10.97%** on Amazon-Beauty).
> *   **Efficiency:** Crucially, RSIR achieves these gains without the heavy overhead of training external generators. As shown in **Table 12** on page 21 (**Appendix D.4**), RSIR is **5$\times$ faster than ASReP** and **18$\times$ faster than DR4SR** during the generation phase.
>
> > **W3 & Q3: Lack of formal theoretical guarantees**
>
> We have expanded **Section 4** and added formal analysis in **Appendix E on page 21**. The recursive error bound specifies the condition under which RSIR remains stable, and we state the assumptions explicitly.
> *   **Formal Proof (Appendix E.1 on page 21):** We mathematically prove that RSIR’s generation process functions as a **Manifold Tangential Gradient Penalty**. This formalizes the "landscape smoothing" claim: we are minimizing the gradient specifically along the directions of the user preference manifold.
> *   **Stability Guarantee (Section 4.2 and Appendix E.2 on page 22):** We derived **Theorem 1 (Recursive Error Bound)**, which proves that the error follows a linear contraction mapping. We identify a specific condition—the **"Breakdown Point"** (Eq. 17)—for the fidelity leakage rate $\tilde{p}_k$. This provides the formal condition you requested: RSIR is guaranteed to help when the fidelity check is strict enough to keep noise below this threshold.
>
> > **W4 & Q4: Missing efficiency reporting**
>
> We have added a comprehensive efficiency analysis in **Section 3.3** and **Appendix D on page 20**.
> *   **Complexity:** We prove the complexity is linear with respect to *vocabulary size $V$ and generation length $L_e$*  (Eq. 9).
> *   **Runtime:** Empirical results (**Table 12** on page 21) confirm that RSIR's retraining time (2m 16s) is comparable to the Base model (2m 34s) and faster than methods like DR4SR (10m+). RSIR is **5$\times$ faster than ASReP** and **18$\times$ faster than DR4SR** during generation.
> *   **Scalability:** We further introduced a **Clustering-based Retrieval strategy (Appendix D.5 on page 20)**, which reduces the fidelity check cost from linear to sub-linear (logarithmic time), ensuring scalability for massive datasets.

---

### Official Review · Reviewer_hwWB · 2025-11-01

**Soundness:** 3
**Presentation:** 3
**Contribution:** 2
**Rating:** 4
**Confidence:** 4

**Summary:**

The authors introduce RSIR (Recursive Self-Improving Recommendation), a framework that enables recommender models to iteratively generate, filter, and retrain on their own synthetic data—improving performance without external supervision.

Each iteration involves: (1) generating user–item sequences, (2) filtering them via a fidelity-based control mechanism to ensure alignment with real user behavior, and (3) retraining a new model on the enriched dataset. Theoretically, RSIR functions as a data-driven implicit regularizer, smoothing the loss landscape and enhancing generalization. Experiments across multiple datasets and model backbones show consistent, cumulative gains over heuristic data augmentation methods such as reordering and insertion.

**Strengths:**

1. RSIR is the first recursive self-improving framework for recommender systems that does not rely on external models or knowledge.
2. The paper provides a theoretical interpretation of RSIR, showing it acts as an implicit regularizer.
3. The experiments are extensive, covering multiple datasets and backbone models, with ablation studies on fidelity control, exploration balance, and hyperparameter sensitivity.

**Weaknesses:**

1. The paper does not clearly explain why existing self-improving methods based on external models (e.g., LLMs) or knowledge are not included for comparison. Even if such exclusion is justified, it would still be valuable to provide comparative results to quantify the benefits (or trade-offs) of using external knowledge.
2. A possible motivation for avoiding external models is computational efficiency, yet the proposed method appears computationally expensive. First, it involves tuning several sensitive hyperparameters via grid search (Line 303-304), which is resource-intensive. Second, the model is retrained "from scratch" on the updated dataset in each iteration, which is especially costly for large backbones. What about fine-tuning with incremental data rather than retraining? Repeated full retraining is often impractical in real-world settings. In fact, the total optimization time may exceed that of methods relying on external models or knowledge. However, the paper provides no analysis of runtime or computational complexity.
3. The explanation of Figure 3 (model performance vs. τ) appears inaccurate or overstated. The trend does not clearly follow a U-shaped curve; for example, performance at τ=10 is substantially worse than at τ=5. As such, the claim is unconvincing and may require revision.

**Questions:**

1. Please address weakness 1
2. Please address weakness 2
3. Please address weakness 3
4. In Line 219, RSIR filters duplicates. However, doesn't exploitation naturally produce sequences that overlap with historical items? Please clarify how duplicates are defined and handled.

---

> ### Author Response · Authors · 2025-11-22
>
> We sincerely thank you for recognizing RSIR as the **"first recursive self-improving framework"** without external dependencies and for valuing our **theoretical contributions**. Your critique regarding the comparison with external models and computational efficiency highlighted the most critical areas for improvement.
>
> To address these concerns, we have significantly revised the paper. We added **fine-tuning experiments (Table 1)**, a **compatibility study with external knowledge (Appendix C.6 on page 18)**, and a rigorous **computational efficiency (Section 3.3)** and **runtime analysis (Appendix D on page 18)**.
>
> > **W1 & Q1: Comparison with External Knowledge**
>
> We agree that understanding RSIR’s position relative to external-knowledge methods is crucial. We have added **Appendix C.6** and **Table 10 on page 18** to address this.
> *   **Orthogonality:** Our research question—whether a model can bootstrap its **own** performance using intrinsic signals—is orthogonal to external knowledge methods.
> *   **Empirical Gain:** To evaluate this, we applied RSIR on top of a **Semantic ID-based model**, which incorporates external semantic hierarchies. As shown in **Table 10 on page 18**, while the Semantic ID baseline is strong (NDCG@10 0.0507 vs. SASRec 0.0477), applying RSIR yields a further **+4.89% gain in Recall@20**. This proves that even models enriched with external knowledge benefit significantly from self-improving.
>
> > **W2 & Q2: Computational Efficiency & Practicality**
>
> Thank you for raising this critical point. We have added comprehensive analysis and new experiments to demonstrate that RSIR is computationally efficient and scalable.
>
> **1. Retraining vs. Fine-Tuning**
> *   You correctly noted that full retraining can be costly. We have added **RSIR-FT (Fine-Tuning)** results to **Table 1**.
> *   **Result:** RSIR-FT achieves performance **comparable to—and in some cases superior to—full retraining** (e.g., **0.0594** Recall@10 for RSIR-FT vs. 0.0578 for RSIR on Amazon-Beauty).
> *   **Implication:** This confirms that practitioners can apply RSIR via incremental fine-tuning, making it highly practical for real-world systems.
>
> **2. Runtime Efficiency & Complexity (Appendix D on page 19)**
> *   **Theoretical Linearity:** We derive that the total complexity is **linear** with respect to the *vocabulary size $V$ and generation length $L_e$* (Eq. 9). Crucially, our "Break" mechanism acts as an early-stopping regularizer, keeping the effective generation length $L_e \ll L$, minimizing overhead.
> *   **Runtime Efficiency:** We explicitly measured runtime against other data-centric baselines in **Table 12 (Appendix D.4)** (page 20, 21). RSIR is **~5x faster than ASReP** and **~18x faster than DR4SR**. This is because RSIR utilizes the lightweight backbone itself for generation, avoiding the heavy inference costs of training external generative models. Besides, RSIR converges in **2m 16s** during training (vs. **2m 34s for the Base model** and 10m40s for DR4SR) under the same hardware conditions.
> *   **Scalability:** For large vocabularies, we introduced a **Clustering-based Retrieval** strategy (**Appendix D.5 on page 20**), which reduces the fidelity check complexity from linear to sub-linear while maintaining performance (**Table 11 on page 21**).
>
> **3. Hyperparameters:**
> *   We derived a theoretical **"Breakdown Point"** (Theorem 1 in **Appendix E.2 on page 22**) that bounds the fidelity leakage rate. This theory narrows the search space substantially. Empirically, we found that optimal hyperparameters are stable across datasets (e.g., $\tau \in [5, 20]$), mitigating the need for the extensive grid search performed in our initial experiments.
>
> > **W3 & Q3: Interpretation of Figure 3**
>
> We appreciate the correction. We have revised **Section 5.3.1** to more accurately describe the trend as a trade-off rather than a simple U-shape.
> *   **Interpretation:** Overly strict thresholds ($\tau \to 1$) choke exploration (low diversity), while overly permissive thresholds ($\tau \to \infty$) introduce noise. The performance peaks in the middle range, consistent with our theoretical finding that we must balance "Expansion" (diversity) with "Contraction" (fidelity).
>
> > **Q4: Definition and Handling of Duplicates**
>
> *   **Definition:** A generated sequence is discarded only if it is an **exact duplicate** (both items and order) of an existing sequence in the training set.
> *   **Exploitation vs. Duplication:** While "Exploitation" samples items from the user's history, the **order** is generated by the model's current probability distribution. RSIR often finds *novel sequential patterns* or high-order correlations among historical items that were not present in the original chronological sequence. Thus, they are not duplicates. Furthermore, it probabilistically switches between Exploitation (history) and Exploration (global items) for each item in a sequence. In our experiments, exact duplicates were extremely rare (<0.03%).

---

### Official Review · Reviewer_tgHh · 2025-11-01

**Soundness:** 3
**Presentation:** 3
**Contribution:** 3
**Rating:** 6
**Confidence:** 3

**Summary:**

The paper proposes the Recursive Self-Improving Recommendation (RSIR) framework, which addresses the challenge of data scarcity in recommender systems by enabling models to bootstrap their performance through iterative self-improvement. In each iteration, the model generates plausible user interactions, filters them for consistency with true user preferences via a fidelity-based quality control mechanism, and retrains itself using the enriched dataset. The paper demonstrates the theoretical advantages of RSIR as an implicit regularizer and provides empirical evidence that the framework improves performance across various datasets and architectures. Notably, even weak models benefit from this self-improvement process.

**Strengths:**

- The introduction of a recursive self-improvement loop in the context of recommender systems is a novel idea and a creative approach to tackling data sparsity. The idea of bootstrapping performance through self-generated data without the need for external models or data is a significant innovation.

- The experiments are thorough and cover multiple benchmark datasets. The use of multiple backbone models (SASRec, CL4SRec, HSTU) demonstrates the framework’s versatility, and the comparisons against heuristic-based augmentation methods provide strong evidence of RSIR’s effectiveness.

- The theoretical analysis of RSIR as an implicit regularizer adds depth to the work and makes the proposed framework more compelling.

**Weaknesses:**

- While the recursive loop is theoretically sound, the practical implementation of such a system might face challenges, especially regarding error amplification in early iterations. The paper discusses the importance of fidelity control, but there may be further considerations regarding the computational complexity and convergence behavior of the loop over many iterations.

- While the fidelity-based quality control mechanism is crucial for preventing performance collapse, a more in-depth discussion of how the parameters (like the rank threshold) are chosen could improve understanding. The paper touches on this, but further details would be helpful for practitioners looking to implement the system.

- While NDCG and Recall are appropriate, additional evaluation metrics such as precision or F1-score could provide a more comprehensive view of the performance, especially in diverse scenarios.

**Questions:**

1. Could the authors elaborate on how the fidelity threshold is selected and whether there are any recommendations for choosing it based on dataset characteristics?

2. How would RSIR behave in cases where there is a high degree of noise in the training data? Would the model still be able to self-correct, or could error propagation become more severe?

3. Are there any limitations in the framework when applied to real-time recommendation systems with dynamic data (e.g., streaming data)?

---

> ### Author Response · Authors · 2025-11-22
>
> We sincerely thank you for your positive evaluation and for recognizing RSIR as a **significant innovation** with a **compelling** theoretical foundation. We appreciate your constructive feedback regarding practical implementation and robustness.
>
> To address your concerns, we have significantly expanded the manuscript with a **formal complexity analysis (Section 3.3, Appendix D on page 19)**, a **noise robustness study (Appendix F on page 23)**, and a **theoretical proof of error convergence (Appendix E on page 21)**.
>
> > **W1: Regarding Complexity and Error Amplification**
>
> *   **Error Amplification (Theoretical Guarantee):** We addressed this by deriving the **Recursive Error Bound (Theorem 1)** in the new **Appendix E.2 on page 22**.
>     *   We identify a **"Breakdown Point" (Eq. 17)**. As long as the fidelity control keeps the leakage rate $\tilde{p}_k$ below this threshold, the system is theoretically explainable to alleviate error explosion. Our Fidelity-based quality control mechanism (Section 3.2.2) is designed precisely to enforce this boundary.
>     *   **Empirical Evidence:** **Figure 2** confirms this analysis, showing that performance improves cumulatively over iterations up to a certain number of iterations without collapse.
>
> *   **Computational Complexity:** We added computational complexity analysis in **Section 3.3** and **Appendix D on page 19**.
>     *   **Linear Scalability:** The total complexity scales linearly with *vocabulary size $V$ and generation length $L_e$* (**Eq. 9**). Because our fidelity check acts as an "early-stopping" mechanism, the effective generation length $L_e$ remains small ($L_e \ll L$), minimizing overhead. Additionally, for large item spaces, we introduced a **Clustering-based Retrieval** strategy (**Appendix D.5 on page 20**), which reduces the fidelity check cost from $O(|\mathcal{V}|)$ to sub-linear, ensuring potential scalability.
>     *   **Runtime Efficiency:** In **Appendix D.4 (Table 12)** on page 21, we demonstrate that RSIR is **5$\times$ faster than ASReP** and **18$\times$ faster than DR4SR** during generation. Under the same hardware conditions, RSIR converges in **2m 16s** during training (vs. 2m 34s for the Base model and 10m40s for DR4SR).
>
> > **W2 & Q1: Selection of the fidelity threshold $\tau$**
>
> *   **Theoretical View:** According to our stability condition (Eq. 17 in Appendix E.2 on page 22), $\tau$ must be strict enough to keep the "noise leakage" $\tilde{p}_k$ lower than the contraction gain. For weaker base models or noisier data, a stricter threshold (e.g., $\tau=5$) is safer to prevent performance collapse.
> *   **Practical Selection:** In our experiments, we used grid search. Empirically, we found that optimal results are consistently achieved with a relatively small threshold (e.g., $\tau=5$ for Amazon-Toys, $\tau=20$ for others), suggesting the parameter does not require extensive tuning.
>
> > **Q2: Behavior with a high degree of noise**
>
> *   **Robustness Test:** We conducted a noisy test in the new **Appendix F** on page 23, injecting up to 80% random noise into the training data.
> *   **Result:** As shown in **Figure 7** on page 23 and **Table 13** on page 24, RSIR maintains superiority over the base model. Notably, the **relative improvement actually increases** as the data becomes noisier (e.g., on Yelp, improvement rises from **+7.55%** at 0% noise to **+16.42%** at 80% noise).
> *   **Why it works:** This validates our **Insight 1 (Section 4.1)**: RSIR acts as a **"Denoising Filter."** Since random noise typically lies off the user preference manifold, our fidelity check rejects it, effectively cleaning the dataset while densifying valid signals.
>
> > **W3: Additional Evaluation Metrics**
>
> *   We have included **Precision, F1-score, and MRR** in **Appendix C.3** on page 16 (**Table 6** on page 17). RSIR achieves consistent, statistically significant gains across all metrics (e.g., +11.1% MRR improvement on Yelp), confirming that the Recall gains do not come at the cost of Precision.
>
> > **Q3: Limitations in real-time/streaming systems**
>
> *   **Inference:** RSIR optimizes the backbone model directly. Therefore, **online inference speed is identical** to the base model, making it potentially suitable for real-time serving.
> *   **Dynamic Training:** Currently, RSIR is designed for periodic offline retraining and does not yet support incremental updates for streaming scenarios. While we discuss strategies to accelerate this process in **Appendix D.5** on page 20, extending the recursive loop to support real-time incremental updates remains a valuable direction for future research.

---

### Author Response · Authors · 2025-12-01
**General Response: Summary of Revisions.**

Dear Reviewers,

Thank you for your thoughtful and constructive feedback during the review process. We are encouraged that you recognized the **novelty** of our recursive self-improving framework (tgHh, Rqbe), the **extensive experimental results** (tgHh, hwWB, Rqbe), the contribution of our **theoretical analysis** (tgHh, hwWB), and the potential of our approach to solve the critical issue of **data sparsity** without reliance on external models (hwWB, Rqbe, jDuY). Your valuable suggestions have significantly strengthened our submission.

---

Dear (New) Area Chair,

We appreciate your effort in taking over the assessment of our submission. Since the original reviewers cannot update their scores, we provide this summary to highlight how we have **resolved the primary concerns** raised in the initial reviews. All changes are highlighted in blue in the **revised PDF**.

**1. Strengthened Theoretical Foundation (Section 4 & Appendix E)**

>Reviewer Rqbe noted that our initial theory was qualitative and requested formal bounds to clarify **"when RSIR helps versus hurts."** Reviewer tgHh also raised concerns about **error amplification** in recursive loops. In response, we added:

*   **Geometric Interpretation (Formal Proof):** We provided a formal proof (Appendix E.1) showing that RSIR functions as a **Manifold Tangential Gradient Penalty**, smoothing the landscape along preference manifolds.
*   **Recursive Error Bound:** We identified a theoretical **"Breakdown Point"** (Theorem 1, Appendix E.2) for the fidelity leakage rate; we prove that as long as the fidelity control keeps noise below this threshold, the system is theoretically guaranteed to prevent error amplification.

**2. Comprehensive Efficiency and Scalability Analysis (Section 3.3 & Appendix D)**

>Reviewers hwWB, Rqbe and jDuY questioned the practicality of the "generate-filter-retrain" loop, specifically asking about **runtime costs** and whether **fine-tuning** could replace full retraining. Reviewer jDuY raised concerns about sampling from **large candidate sets**.

*   **Computational Complexity:** We added a formal analysis in Section 3.3 showing that total complexity scales **linearly** with *vocabulary size $V$ and generation length $L_e$* (Eq. 9).
*   **Runtime Costs:** We added an empirical runtime comparison (Table 12). Results show RSIR generation is **$\approx$5x faster than ASReP** and **$\approx$18x faster than DR4SR**. Counter-intuitively, RSIR training (2m 16s) is faster than the base model (2m 34s) due to accelerated convergence.
*   **Scalability on Large Item Sets:** We introduced a Clustering-based Approximate Retrieval strategy (Appendix D.5), reducing the fidelity check complexity from linear $O(|V|)$ to **sub-linear (logarithmic)** while maintaining performance, ensuring scalability for massive item sets.
*   **Viability of Fine-Tuning:** We added **RSIR-FT** results (Table 1), demonstrating that incremental fine-tuning achieves **comparable performance** to full retraining, making the framework highly practical for real-world deployment.

**3. Expanded Baselines and Metrics (Table 1 & Appendix C.3)**

>Reviewer Rqbe noted that the **baseline** coverage was narrow, and tgHh suggested more comprehensive metrics.

*   **Generative Baselines:** We added comparisons against three state-of-the-art generative augmentation methods: **ASReP** (sequence extension), **DiffuASR** (diffusion-based), and **DR4SR** (regeneration). RSIR consistently outperforms these methods while being significantly more efficient.
*   **Additional Metrics:** We expanded evaluation to include **Precision, F1-score, and MRR** (Table 6), showing consistent gains across all dimensions of recommendation quality.

**4. Robustness and Compatibility (Appendix F & C.6)**

>Reviewer tgHh asked about behavior under **high noise**, and hwWB questioned the relationship with **external knowledge**.

*   **Noise Robustness:** We conducted a stress test with up to 80% injected noise (Appendix F). RSIR demonstrates superior robustness, with relative gains **increasing** as data becomes noisier, validating its role as a "denoising filter."
*   **External Knowledge:** We demonstrated that RSIR is **orthogonal** to external knowledge methods (Appendix C.6), providing additive gains (**+4.89%**) when applied on top of a Semantic ID-based model.

We believe these revisions address the reviewers' concerns and highlight the theoretical soundness and practical value of RSIR. We are happy to clarify any remaining questions in the final days of the process.

Best regards,

The Authors

---

> ### Author Response · Authors · 2025-12-01
> **Summary of Changes in the Revised PDF**
>
> To AC and Reviewers,
>
> To facilitate your review, we list the specific sections added or modified in the revised PDF:
>
> *   **Section 3.3 (New):** Added a formal Computational Complexity Analysis proving linear scalability  (**Related to Point 2**).
> *   **Section 4.1 (Updated) & Section 4.2 (New) & Appendix E (New):** Added formal Theoretical Analysis, including the derivation of the Implicit Regularizer (Manifold Tangential Gradient Penalty) and the Recursive Error Bound / Convergence Proof (**Related to Point 1**).
> *   **Table 1 (Updated):** Integrated results for three new generative baselines (**ASReP, DiffuASR, DR4SR**) and the Fine-Tuning variant (**RSIR-FT**) (**Related to Points 2 & 3**).
> *   **Appendix C.3 (New):** Added evaluation on Precision, F1-score, and MRR (Table 6) (**Related to Point 3**).
> *   **Appendix C.6 (New):** Added compatibility analysis with External Knowledge (Semantic IDs) (Table 10) (**Related to Point 4**).
> *   **Appendix D (New):** Detailed Computational Complexity derivation, Empirical Runtime Analysis (Table 12), and Scalability Optimization via Clustering (Table 11) (**Related to Point 2**).
> *   **Appendix F (New):** Robustness Analysis under varying levels of data noise (Figure 7, Table 13) (**Related to Point 4**).
>
> Best regards,
>
> The Authors

---

### Meta-Review · Area_Chair_mrxp · 2025-12-25

**Summary:**

This paper proposes the RSIR framework for recursive self-improving recommendation to address data sparsity, with a core loop of generating synthetic user interaction sequences, filtering via fidelity control, and retraining. The reviewers’ key concerns centered on theoretical rigor, computational efficiency and scalability in practical scenarios, baseline coverage comprehensiveness, reliance of the fidelity control mechanism on future ground truth, and limitations in dynamic real-time systems. After synthesizing the reviewers’ evaluations and the authors’ rebuttal, the suggested decision is to reject this submission.

**Reviewer Concerns:**

The authors’ rebuttal has effectively addressed several reviewer concerns: they supplemented formal theoretical analyses including manifold tangential gradient penalty and recursive error bounds to strengthen the theoretical foundation, provided detailed computational complexity derivation, runtime comparisons, and scalability optimizations (e.g., clustering-based retrieval) to respond to efficiency questions, expanded generative baselines (e.g., ASReP, DiffuASR) and additional evaluation metrics (Precision, F1-score, MRR) to enhance baseline coverage, and verified the feasibility of fine-tuning as an alternative to full retraining for practical deployment. However, some core concerns remain outstanding: the fidelity control mechanism’s inherent reliance on future ground truth, which may limit generalization to truly novel user trajectories, has not been fundamentally resolved; the framework’s adaptability to real-time streaming data with dynamic updates remains unaddressed as a key practical limitation; and while theoretical bounds were added, some reviewers’ expectations for deeper insight into the framework’s behavior under extreme conditions (e.g., highly sparse or dynamic data) were not fully met.

**Reviewer Scores:**

The initial reviewer scores were bellow the acceptance threshold, reflecting recognition of the work’s novelty but concerns about its rigor and practicality. Following the authors’ rebuttal, which addressed several technical questions and supplemented key experiments, the reviewers would likely have modestly adjusted their scores upward due to the strengthened theoretical and empirical support. However, given the unresolved core limitations related to generalization, real-time applicability, and the fidelity control mechanism’s dependency, the overall scores would still fall below the conference’s acceptance threshold. It is worth noting that the research theme of recursive self-improvement for recommendation systems is meaningful and addresses a critical challenge of data sparsity. The authors are encouraged to further refine the framework by addressing the outstanding concerns, particularly optimizing the fidelity control mechanism to reduce reliance on future ground truth and extending support for dynamic streaming scenarios.

---

### Decision · Program_Chairs · 2026-01-26

Reject